# MAF1 represses *CDKN1A* through a Pol III-dependent mechanism

**Yu-Ling Lee, Yuan-Ching Li, Chia-Hsin Su, Chun-Hui Chiao, I-Hsuan Lin, Ming-Ta Hsu***

Institute of Biochemistry and Molecular Biology, School of Life Science, National Yang-Ming University, Taipei, Taiwan

**Abstract** MAF1 represses Pol III-mediated transcription by interfering with TFIIIB and Pol III. Herein, we found that *MAF1* knockdown induced *CDKN1A* transcription and chromatin looping concurrently with Pol III recruitment. Simultaneous knockdown of *MAF1* with Pol III or *BRF1* (subunit of TFIIIB) diminished the activation and looping effect, which indicates that recruiting Pol III was required for activation of Pol II-mediated transcription and chromatin looping. Chromatin-immunoprecipitation analysis after *MAF1* knockdown indicated enhanced binding of Pol III and BRF1, as well as of CFP1, p300, and PCAF, which are factors that mediate active histone marks, along with the binding of TATA binding protein (TBP) and POLR2E to the *CDKN1A* promoter. Simultaneous knockdown with Pol III abolished these regulatory events. Similar results were obtained for *GDF15*. Our results reveal a novel mechanism by which MAF1 and Pol III regulate the activity of a protein-coding gene transcribed by Pol II.

## Introduction

Transcription by RNA polymerase III (Pol III) is regulated by MAF1, which is a highly conserved protein in eukaryotes (*Pluta et al., 2001*; *Reina et al., 2006*). MAF1 represses Pol III transcription through association with BRF1, a subunit of initiation factor TFIIIB, which prevents attachment of TFIIIB onto DNA. This interaction also inhibits Pol III from binding to BRF1, which in turn prevents recruitment of Pol III to Pol III promoters. Furthermore, MAF1 also inhibits Pol III transcription through direct binding with Pol III, which interferes with the recruitment of Pol III to the assembled TFIIIB/DNA complexes (*Desai et al., 2005*; *Vannini et al., 2010*). In addition, association of MAF1 with Pol III-transcribed genes has been detected genome-wide concomitant with an increase in occupation during repression; this indicates that direct interaction of MAF1 with Pol III genes is also an important attribute of repression (*Roberts et al., 2006*).

MAF1 has also been proposed to have the potential to repress Pol II-mediated transcription via repression of *TBP* transcription due to binding of MAF1 to the Elk-1 site on the *TBP* promoter (*Johnson et al., 2007*). Thus, to investigate the potential regulatory role of MAF1 in Pol II genes, we carried out *MAF1* knockdown coupled with microarray analysis. Microarray analysis showed that 124 genes were upregulated and 170 genes were downregulated more than twofold after *MAF1* knockdown. Ingenuity Pathway Analysis (IPA) indicated that most of these genes are related to cell proliferation. Among them, *CDKN1A* (also known as p21) was significantly upregulated and the mechanism of induced transcription of this gene after *MAF1* knockdown was further investigated.

*CDKN1A* is a cyclin-dependent kinase inhibitor that inhibits cell cycle progression through interaction with cyclins and cyclin-dependent kinases (CDKs). As a member of the Cip and Kip family of CDK inhibitors, *CDKN1A* mediates p53-dependent cell-cycle arrest at the $G_1$ phase by inhibiting the activity of *CDK2* and *CDK1* (also known as *CDC2*). In addition, *CDKN1A* also inhibits the activity of proliferating cell nuclear antigen and blocks DNA synthesis and repair as well as cell-cycle

*For correspondence: mth@ym.edu.tw

**Competing interests:** The authors declare that no competing interests exist.

**eLife digest** An organism's genetic material is made of segments of DNA called genes, which contain instructions to make proteins. First, copies of the DNA are made using another molecule called ribonucleic acid (RNA) in a process known as transcription. Then the RNA is used as a template to make a protein. During transcription, enzymes called RNA polymerases move along the DNA to produce the RNA copies.

When a cell is actively growing it needs large quantities of new proteins to be made, and so the level of transcription is higher. However, if a cell experiences stress caused by adverse environmental conditions (e.g., high temperatures), it can conserve resources by shutting down transcription. For example, one RNA polymerase—called Pol III—makes RNA copies with the help of a protein called BRF1 and several other proteins. However, when a cell is under stress, another protein called MAF1 can interfere with transcription by binding to BRF1, which prevents it from interacting with Pol III.

Previous work has suggested that MAF1 can also inhibit the activity of another RNA polymerase called Pol II, but it was not clear how this could work. Lee et al. studied the effect of MAF1 on transcription in human cells. The experiments show that MAF1 blocks the transcription of many genes that are transcribed by Pol II, including one called *CDKN1A*.

*CDKN1A* is involved in regulating many important processes, including the growth of cells and cell death. Cells that produced lower amounts of MAF1 had higher levels of *CDKN1A* transcription, and several proteins—including Pol II, Pol III and BRF1—were more able to bind to this gene. However, this effect was not observed in cells that also produced lower levels of Pol III or BRF1, suggesting that Pol III is needed for Pol II to be able to transcribe *CDKN1A*.

Taken together, Lee et al.'s findings suggest that MAF1 inhibits the transcription of *CDKN1A*, and possibly other genes transcribed by Pol II, by regulating the activity of Pol III. Further research is needed to understand the details of how this works.

progression. As a result, *CDKN1A* can regulate many cellular processes, such as proliferation, differentiation, apoptosis, metastasis, cell survival, and stem cell renewal. Expression of *CDKN1A* can be regulated at the transcriptional level by oncogenes and tumor suppressor proteins that bind various transcription factors to specific elements in response to a variety of intracellular and extracellular signals (*Abbas and Dutta, 2009*; *Warfel and El-Deiry, 2013*).

In this study, we showed that MAF1 can bind to the *CDKN1A* promoter to repress its transcription. Enhanced binding of Pol III after *MAF1* knockdown induced *CDKN1A* transcription and chromatin looping by recruiting common Pol II and Pol III transcription factors as well as binding of TBP, p300, CFP1, and PCAF, along with increase in histone modifications associated with gene activation. Simultaneous knockdown of Pol III and *MAF1* abolished both promoter looping and activation of *CDKN1A* transcription, which indicates that Pol III actively participated in regulation of Pol II genes. Similar results were observed in another cell proliferation-related gene, *GDF15*. These observations reveal a new type of gene regulation in which binding of MAF1 regulates Pol III-mediated transcriptional activation and chromatin looping of Pol II genes.

## Results

### MAF1 knockdown strongly upregulated *CDKN1A* expression

To examine whether MAF1 has the potential to repress Pol II-transcribed genes, we first examined the knockdown effect of *MAF1* by quantitative RT-PCR (qRT-PCR) and immunoblot using multiple siRNAs (*Figure 1A,B*). The siRNA with the strongest knockdown effect was used to perform expression analysis using microarray. 124 Pol II-transcribed genes were upregulated more than twofold after *MAF1* knockdown. Among them, *CDKN1A* was significantly upregulated, resulting in the down-regulation of positive cell cycle regulators. Consistent with expression data, flow cytometry analysis showed that *MAF1* knockdown arrested cells at the $G_1$ phase (*Figure 1C*). We carried out qRT-PCR to confirm whether *CDKN1A* expression was upregulated by *MAF1* knockdown. Efficiency of *MAF1* knockdown was verified by the strong upregulation of two products of Pol III, pretRNA$^{Tyr}$ and pretRNA$^{Leu}$ (*Reina et al., 2006*) (*Figure 1D*). Consistent with microarray analysis, qRT-PCR and

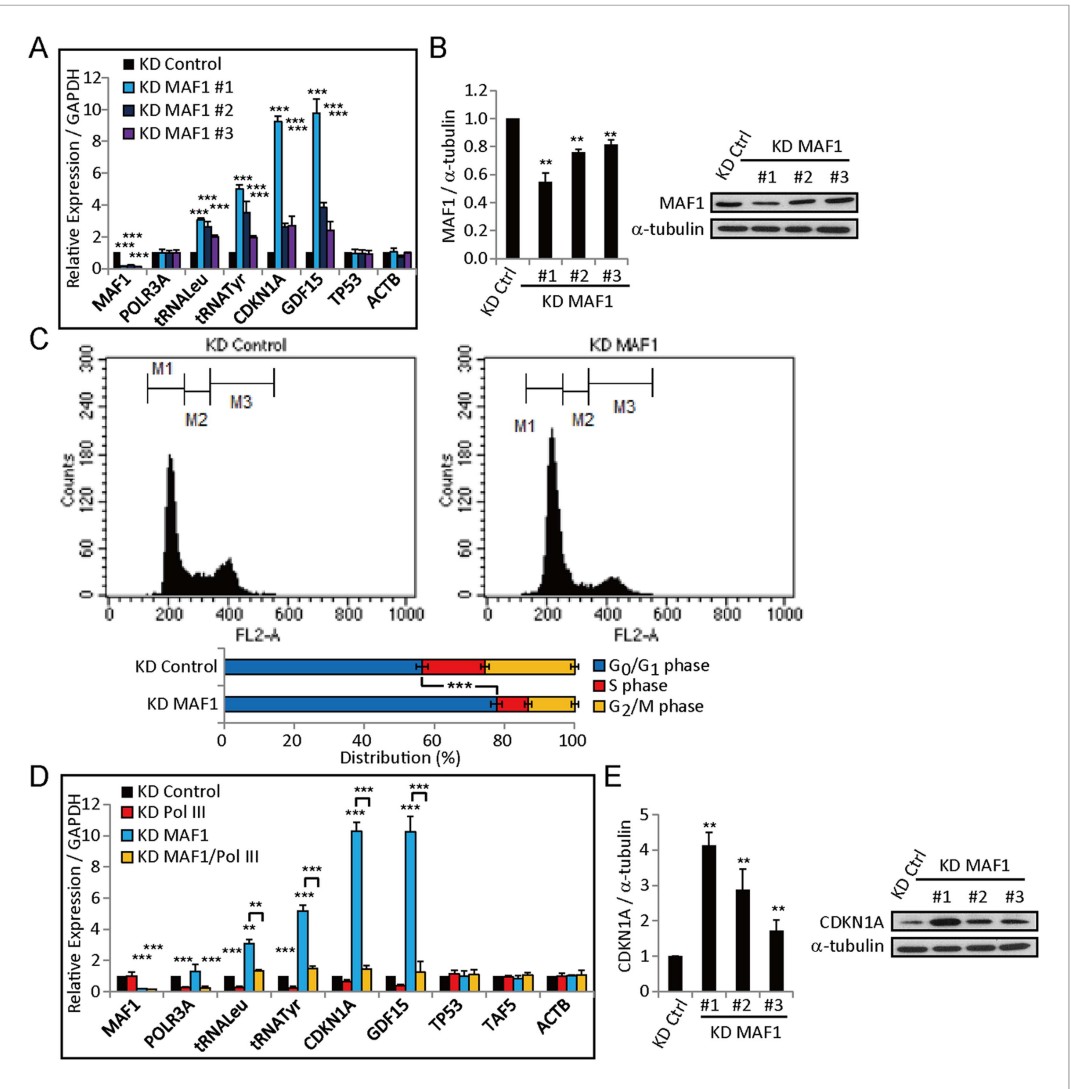

**Figure 1**. *MAF1* knockdown strongly upregulates *CDKN1A* expression and arrests MCF-7 cells at the $G_0/G_1$ phase. Analysis of *MAF1* expression after *MAF1* knockdown using three different siRNAs in MCF-7 cells by quantitative RT-PCR (**A**) and immunoblot analysis (**B**). The immunoblot results were quantified (left panel) using α-tubulin as a loading control on a representative gel (right panel). (**C**) *MAF1* knockdown arrested the MCF-7 cell cycle at the $G_0/G_1$ phase. At 72 hr after knockdown, cells were stained with propidium iodide and subjected to cell cycle analysis by flow cytometry (top panel). The quantification results show that *MAF1* knockdown increased cells arrested at the $G_0/G_1$ phase by 16.4% ± 1.76% (bottom panel). (**D**) Quantitative RT-PCR of genes in MCF-7 cells subjected to siRNA knockdown for 72 hr. *CDKN1A* expression was upregulated 10-fold, and upregulation was abolished by double knockdown of *MAF1* and *POLR3A*. Relative expression normalized to *GAPDH* is displayed. (**E**) Immunoblot analysis of *CDKN1A* expression after *MAF1* knockdown in MCF-7 cells. The results were quantified (left panel) using α-tubulin as a loading control on a representative gel (right panel). All data shown represent mean ± SD, $n \geq 3$, **$p < 0.01$, ***$p < 0.001$ (*t*-test).

The following figure supplement is available for figure 1:

**Figure supplement 1**. *MAF1* knockdown upregulates *CDKN1A* and *GDF15* expression in HCT116^[p53+/+] (wild-type), HCT116^[p53−/−] (p53-null), MCF-10A, and MDA-MB-231 cell lines.

immunoblot analysis showed that *CDKN1A* expression was upregulated about 10-fold after *MAF1* knockdown (*Figure 1D,E*). *GAPDH*, *ACTB*, and *TAF5*, genes that were not affected by *MAF1* knockdown in the microarray, were chosen as the control for qRT-PCR. Expression of these genes was not affected by *MAF1* knockdown (*Figure 1D*).

Because *CDKN1A* is a downstream target of p53 (*Allen et al., 2014*), we further performed *MAF1* knockdown in HCT116^{p53+/+} (wild-type) and HCT116^{p53−/−} (p53-null) cell lines to analyze whether the induced *CDKN1A* expression is dependent on p53. The absence of p53 in HCT116^{53−/−} was confirmed by immunoblot (*Figure 1—figure supplement 1A*). *CDKN1A* expression was induced after *MAF1* knockdown in both wild-type and p53-null HCT116, which indicates that the activation is independent of p53 (*Figure 1—figure supplement 1B,C*). Immunoblot analysis also showed that *CDKN1A* protein level was upregulated after *MAF1* knockdown in p53-null HCT116 (*Figure 1—figure supplement 1D*). *CDKN1A* activation by *MAF1* knockdown was also found in a non-tumorigenic cell line (MCF-10A) and a p53 mutant breast cancer cell line (MDA-MB-231) (*Figure 1—figure supplement 1E,F*). Together, these results demonstrate that MAF1 can regulate *CDKN1A* expression in a variety of cell types independent of p53.

*CDKN1A* activation after *MAF1* knockdown could be due either to interference of binding of transcription factors to the *CDKN1A* promoter by MAF1 or to the active recruitment or activation of Pol III after *MAF1* knockdown. To determine which of these two mechanisms are involved in this process, we carried out simultaneous knockdown of both *MAF1* and Pol III. The former mechanism would not be affected by the double knockdown, whereas the latter would be. The effect of Pol III knockdown was analyzed by qRT-PCR and immunoblot using multiple siRNA sequences (*Figure 1—figure supplement 1G,H*). Simultaneous knockdown of Pol III and *MAF1* indeed abolished the induction of *CDKN1A* expression by knockdown of only *MAF1* (*Figure 1D* and *Figure 1—figure supplement 1B,C,E,F*) in five different cell lines. A control experiment using two Pol III genes, pretRNA^{Tyr} and pretRNA^{Leu}, confirmed the efficiency of Pol III knockdown (*Figure 1D* and *Figure 1—figure supplement 1B,C,E,F*). Knockdown of Pol III alone did not significantly affect *CDKN1A* expression (*Figure 1D*). This result indicates that Pol III plays a critical role in activation of *CDKN1A* expression by *MAF1* knockdown.

Because mRNA levels can be affected by transcription, post-transcriptional processing as well as RNA turnover rate, upregulation of gene expression after *MAF1* knockdown could be due to post-transcriptional mechanisms other than transcription activation. To demonstrate that the induced *CDKN1A* expression indeed occurs at the transcriptional level, we analyzed the rate of nascent transcription by conducting a nuclear run-on experiment after *MAF1* knockdown or simultaneous knockdown of *MAF1* and Pol III. The run-on nascent RNA was labeled with biotin, affinity purified, and analyzed by RT-PCR. A negative control without biotin labeling was used. Consistent with qRT-PCR analysis, nascent transcription of *CDKN1A* was indeed induced after *MAF1* knockdown, whereas transcription of *ACTB* and *TAF5* was not affected (*Figure 2A,B*). Simultaneous knockdown of *MAF1* and Pol III diminished the induced nascent RNA transcription of *CDKN1A* by knockdown of *MAF1* alone (*Figure 2A,B*). These results indicate that Pol III is required for the induction of *CDKN1A* upon MAF1 knockdown.

## R-looping analysis indicated that expression of *CDKN1A* is regulated by MAF1 and Pol III at the transcriptional level

Recent evidence indicated that R-loop formation positively correlates with active transcription in human cells by maintaining the unmethylated state at promoters with skewed guanine-cytosine (GC) content (*Ginno et al., 2012*). The high GC skew of the *CDKN1A* promoter prompted us to test whether the R-loop was present in this region during the activation of transcription by *MAF1* knockdown (*Figure 2C*). We performed R-loop foot-printing by native sodium bisulfite treatment, which converts cytosine to uracil only on the single-stranded DNA (*Yu et al., 2003*). *MAF1* knockdown resulted in the formation of the extended R-loop in the gene body of *CDKN1A*, which indicates active transcription, whereas simultaneous knockdown of Pol III and *MAF1* inhibited R-loop formation (*Figure 2C–E*). The R-looping of the control gene *ACTB*, an active housekeeping gene with high GC skew and expression, was not affected by *MAF1* knockdown (*Figure 2F–H*). The results of R-loop formation and nuclear run-on described above further confirmed that the upregulation of *CDKN1A* expression by *MAF1* knockdown and recruitment of Pol III occurred at the transcriptional level.

## *MAF1* knockdown enhanced binding of Pol III and Pol II at the *CDKN1A* promoter

Expression analysis indicated that *CDKN1A* expression is strongly upregulated after *MAF1* knockdown, and simultaneous knockdown of *MAF1* and Pol III diminished the induced expression.

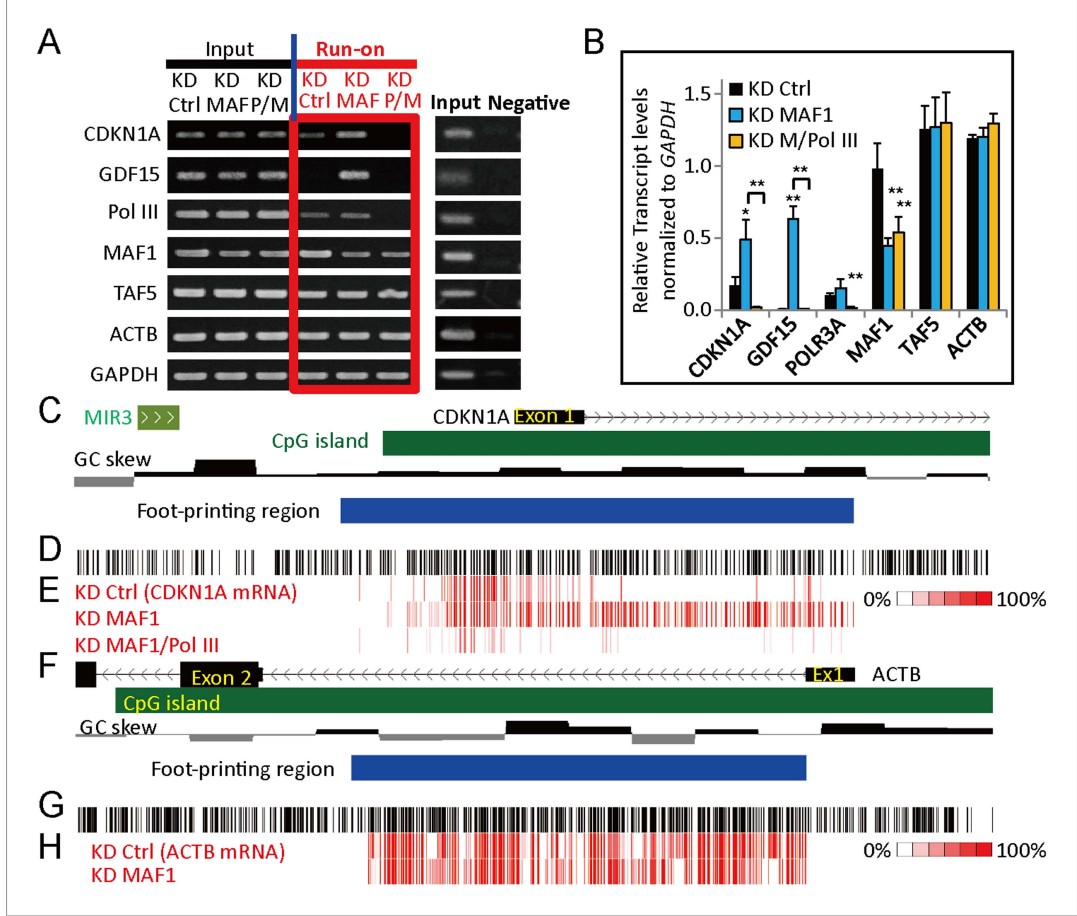

**Figure 2**. *MAF1* knockdown upregulates *CDKN1A* at the transcriptional level. (**A**) For run-on assay, MCF-7 cells were subjected to siRNA knockdown of *MAF1* (KD MAF) or simultaneous knockdown of *MAF1* and Pol III for 72 hr (KD P/M). Nuclei were prepared, and a run-on reaction was performed. Run-on biotin-labeled newly transcribed RNA (Run-on) was affinity purified and subjected to RT-PCR (left panel). Input indicates total RNA before affinity purification, and a negative control was performed by omitting biotinylated nucleotides and subjected to RT-PCR (right panel). (**B**) The run-on results were quantified, and the data shown represent mean ± SD, n = 3, *p < 0.05, **p < 0.01 (t-test). (**C**) Schematic diagram of the *CDKN1A* promoter, including locations of exon 1 (black rectangle), SINE (MIR3), CpG island (green rectangle), guanine-cytosine (GC) skew, and R-loop foot-printing region (blue rectangle). (**D**) Each vertical black line indicates the position of a cytosine on the sense DNA strand. (**E**) Analysis of R-loop foot-printing was performed by native sodium bisulfite treatment followed by PCR amplification and cloning. A total of at least 10 clones were obtained for each knockdown condition (knockdown control, 'KD Ctrl'; knockdown *MAF1*, 'KD MAF1'; and simultaneous knockdown of *MAF1* and Pol III, 'KD MAF1/Pol III'). Each vertical red line represents a converted cytosine to thymine in the sense direction (*CDKN1A* mRNA) for the knockdown control, knockdown *MAF1*, and simultaneous knockdown of *MAF1* and Pol III. Percentage indicates how many clones at a particular cytosine were converted. Knockdown *MAF1* extended the length of R-loop formation in *CDKN1A*, whereas simultaneous knockdown of *MAF1* and Pol III abolished the extension. This indicates that regulation of *CDKN1A* expression by MAF1 and Pol III occurs at the transcriptional level. Background conversion (approximately 5% of cytosine) may be seen because of DNA breathing during the prolonged incubation at 37°C in our data and data produced by others (*Yu et al., 2003*). (**F**) Schematic diagram of *ACTB*, including locations of exons, CpG island, GC skew, and R-loop foot-printing region. (**G**) Each vertical black line indicates the position of a cytosine on the sense DNA strand. (**H**) Each vertical red line represents a converted cytosine to thymine in the sense direction (*ACTB* mRNA) for knockdown control and knockdown *MAF1*. Knockdown *MAF1* did not affect the length of R-loop in *ACTB*, which correlates with the expression data from *Figure 1A*.

These results indicate that removal of *MAF1* may induce recruitment of Pol II and Pol III to activate transcription. To examine this possibility, chromatin-immunoprecipitation (ChIP) analysis followed by quantitative PCR (qPCR) was performed under various knockdown conditions. Efficiency of *MAF1* or

Pol III knockdown as well as simultaneous knockdown of *MAF1* and Pol III was verified by the binding of Pol III to pretRNA^Arg and pretRNA^Leu genes. The results showed enhanced binding of Pol III at pretRNA^Arg and pretRNA^Leu after *MAF1* knockdown, and the binding was diminished after double knockdown (*Figure 3A,B*). Examination of *CDKN1A* gene in the UCSC Genome Database shows that there are two transcription start sites, NM_001220777 (long form) and NM_001220778 (short form), which are 2.25 kb apart (*Figure 3—figure supplement 1A*). Although the expression of both forms was induced after *MAF1* knockdown, the short form had higher expression level and stronger promoter activity in the MCF-7 cell line (*Figure 3—figure supplement 1B–D*). ChIP analysis showed that MAF1 was associated with both transcription start site regions (*Figure 3C,D*). Furthermore, *MAF1* knockdown resulted in the depletion of this regulatory factor with concomitant increase in the binding of both Pol II and Pol III polymerases to both transcription start site regions (*Figure 3E,F*).

Consistent with the expression data, there was significant increase in the binding of active, Serine-5-phosphorylated Pol II at the *CDKN1A* promoter after *MAF1* knockdown, which indicates that the gene was in the active transcription state. The binding of active Pol II was abolished after simultaneous knockdown of Pol III and *MAF1* (*Figure 3F*). Simultaneous knockdown of *MAF1* and *BRF1*, a subunit of TFIIIB that associates with Pol III and is required for binding of Pol III to the DNA template, also abolished the enhanced binding of Pol III and Pol II after *MAF1* knockdown (*Figure 3E,F*). ChIP analysis also indicated induced binding of BRF1 after *MAF1* knockdown, whereas the binding was diminished under simultaneous knockdown of either Pol III or *BRF1* with *MAF1* (*Figure 3G*). These results therefore support the mechanism that recruitment of Pol III to the promoter after *MAF1* knockdown enhances *CDKN1A* expression. Expression of *ACTB* and *TAF5* was not affected because MAF1 and Pol III did not bind to their promoters (*Figure 3H,I*) and their expression was not affected by *MAF1* knockdown; therefore, they were used as negative controls. The transcription of these two control genes was not affected by single or double knockdown of Pol III and/or *MAF1* (*Figure 1D*).

## MAF1 binds to the short interspersed element (SINE) of *CDKN1A*

Because ChIP data revealed that MAF1 may directly bind to the promoter, we searched for potential binding sites in the *CDKN1A* promoter region. We noticed a MIR3 element (a SINE with Pol III promoter) that could be transcribed by Pol III and therefore might represent the target-binding site of MAF1. To test this possibility, we used an in vitro DNA binding reaction (*Britten, 1996*; *Toth and Biggin, 2000*) to examine whether purified MAF1 protein could bind to the cloned *CDKN1A* promoter. Consistent with ChIP, in vitro DNA binding assay showed that purified MAF1 protein could indeed bind to the *CDKN1A* promoter that contained the MIR3 element (*Figure 4A,B*), but the binding was abolished when the MIR3 repeat element was deleted, which indicates specificity of MAF1 binding to the MIR3 element (*Figure 4A,B*). To further show that the Pol III promoter of the MIR3 element is responsible for the binding, an in vitro binding assay was carried out with DNA in which the MIR3 DNA sequence from the *CDKN1A* promoter had a deleted or mutated A-box sequence (Pol III promoter element). In vitro DNA-protein binding assays performed by the above method or colorimetric assay (Abcam, ab117139) both showed that the DNA with deleted or mutated A-box sequences exhibited significantly lower binding of MAF1, which indicates the specificity of MAF1 for the Pol III promoter element (*Figure 4A–C*). Moreover, consistent with ChIP data, MAF1 did not bind in vitro to the *ACTB* promoter that did not contain a SINE or sequences that would resemble the Pol III promoter element (*Figure 4A,B*). To the best of our knowledge, this is the first demonstration of direct binding of MAF1 to a specific DNA sequence.

## In vitro transcription using HeLa cell nuclear extract demonstrated that transcription of Pol II genes was reciprocally regulated by MAF1 and Pol III

Although the several types of evidence discussed above strongly support the regulation of *CDKN1A* by recruitment of Pol III to the promoter, the effect observed in vivo nevertheless could be due to some other indirect effect. To directly demonstrate the enhancement of Pol II transcription by removing MAF1 and recruiting Pol III, we carried out in vitro transcription using commercial HeLa cell nuclear extract. Constructed DNA templates of the genes analyzed are described in the 'Materials and methods'. The in vitro, newly transcribed RNA was labeled with biotin, and the products were affinity purified. The nature of the affinity-purified nascent RNA was then analyzed by qRT-PCR.

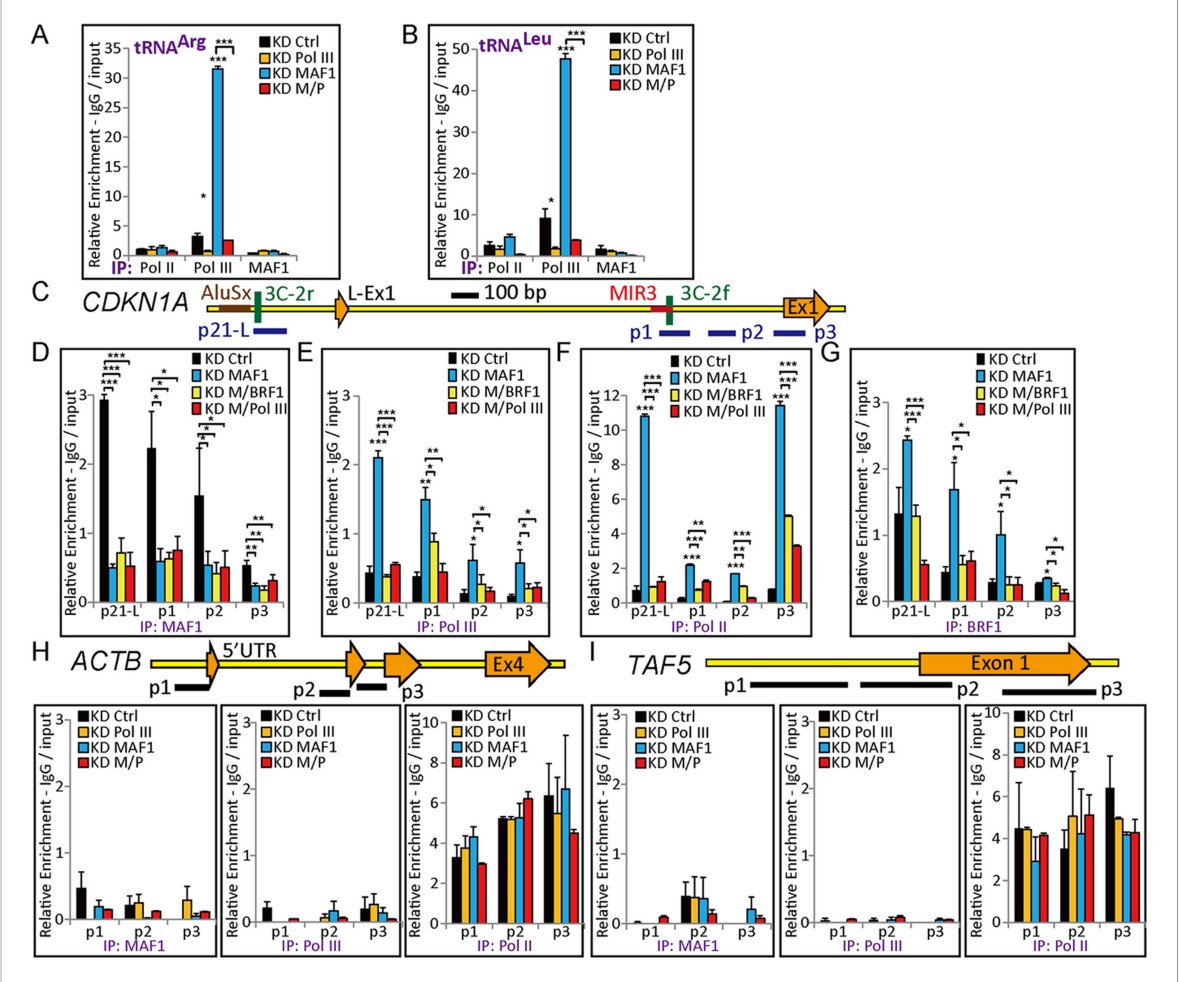

**Figure 3**. *MAF1* knockdown enhanced binding of Pol III and Pol II at the *CDKN1A* promoter. ChIP was performed in MCF-7 cells subjected to siRNA knockdown for 72 hr. DNA isolated from immunoprecipitated chromatin was subjected to qPCR and calculated as indicated in the 'Materials and methods'. Significant binding of Pol III was detected at two tRNA genes, tRNA$^{Arg}$ (**A**) and tRNA$^{Leu}$ (**B**), after *MAF1* knockdown (KD MAF1). The enhance binding of Pol III was diminished when there was simultaneous knockdown of *MAF1* and Pol III (KD M/Pol III). (**C**) Diagram of the *CDKN1A* promoter, including locations of exon 1 (long form: L-Ex1, and short form: Ex1), SINEs (AluSx and MIR3), and ChIP–qPCR amplicons (p21-L, p1, p2, and p3). (**D**) Binding of MAF1 was detected at the *CDKN1A* promoter, which diminished after *MAF1* knockdown. (**E**) Enhanced binding of Pol III was detected at the *CDKN1A* promoter after *MAF1* knockdown. (**F**) *MAF1* knockdown indicates enhanced binding of Serine-5-phosphorylated Pol II, which was abolished when there was simultaneous knockdown of Pol III and *MAF1*. (**G**) Enhanced binding of BRF1 was detected at the *CDKN1A* promoter after *MAF1* knockdown. (**H**) Top panel: diagram of the *ACTB* promoter, including locations of each exon (Ex1 to Ex4) and ChIP–qPCR amplicons (p1, p2, and p3). Bottom panel: neither MAF1 nor Pol III was detected at the *ACTB* promoter. Only binding of Pol II was detected at the *ACTB* promoter. (**I**) Top panel: diagram of the *TAF5* promoter, including locations of exon 1 and ChIP–qPCR amplicons (p1, p2, and p3). Bottom panel: neither MAF1 nor Pol III was detected at the *TAF5* promoter. Only binding of Pol II was detected at the *TAF5* promoter. All data shown represent the mean ± s.e.m., $n \geq 3$, *$p < 0.05$, **$p < 0.01$, ***$p < 0.001$ (*t*-test).

The following figure supplement is available for figure 3:

**Figure supplement 1**. Expression and promoter activity of *CDKN1A* transcripts in the MCF-7 cell line.

Negative control without biotin labeling was used. First, we showed that the in vitro transcription was indeed mediated by Pol II by inhibition of transcription either by α-amanitin treatment (*Figure 4D*) or by depletion of Pol II in the extract using anti-Pol II antibody (*Figure 4E*). When purified MAF1 protein was pre-incubated with DNA template prior to the addition of nuclear extract, *CDKN1A* transcription was repressed with respect to the control (*Figure 4D*). This result is consistent with in vivo expression analysis and the in vitro binding assay. Together these results suggest that MAF1 protein serves as

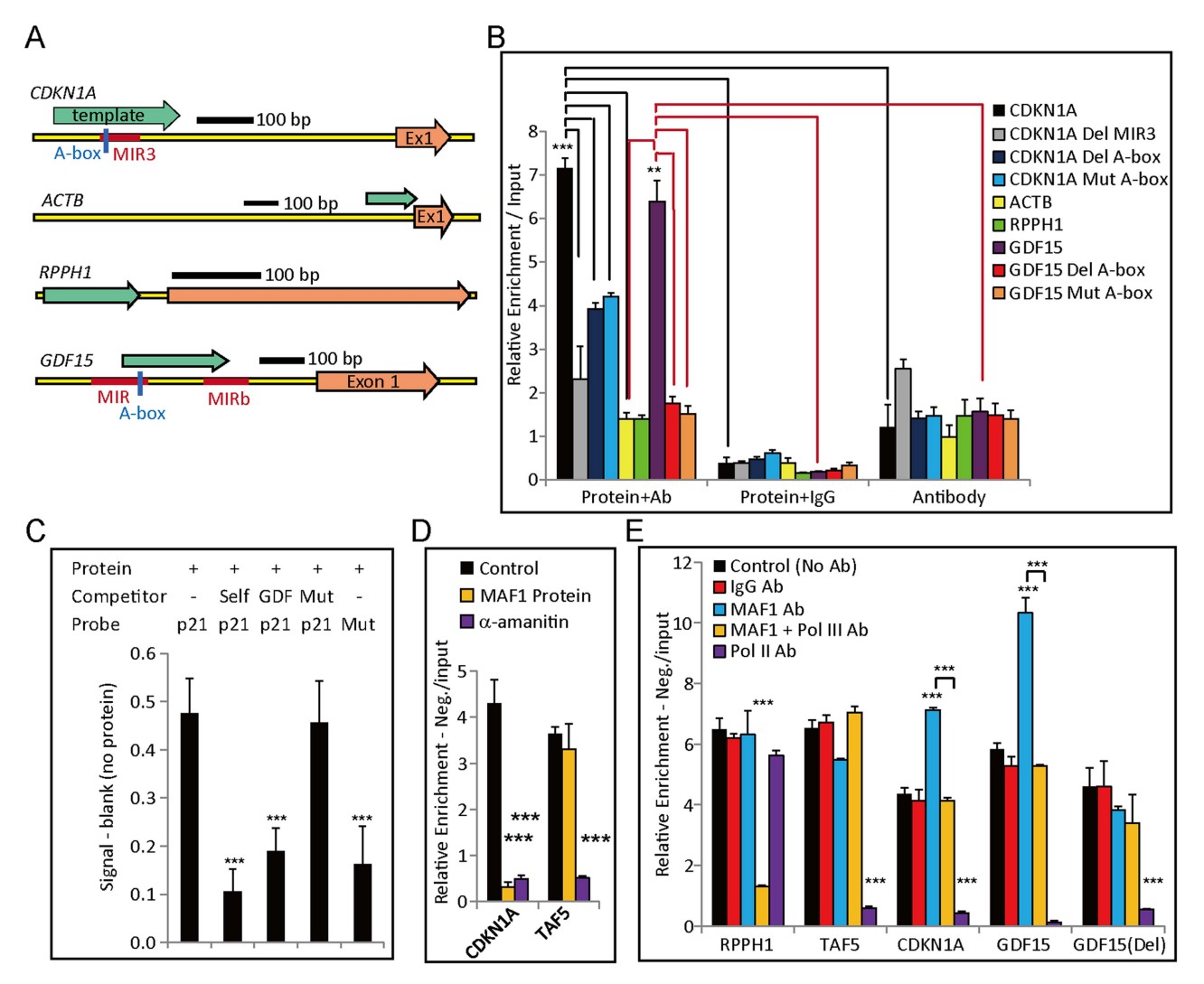

**Figure 4**. In vitro binding and transcription assays demonstrate MAF1-regulated Pol III-mediated activation of Pol II-regulated genes. (**A**) Diagrams of Pol II promoters (*CDKN1A*, *ACTB*, *RPPH1*, and *GDF15*) with locations of exon 1, SINEs (red), and constructed DNA template (green arrow) for the in vitro MAF1 binding assay. (**B**) An in vitro DNA binding assay was performed as described in the 'Materials and methods'. In brief, DNA template, MAF1 protein (His-tagged), and Anti-6× His tag antibody were added to the binding reaction (Protein + Ab). A negative control was performed by substituting IgG antibody for Anti-6× His tag antibody (Protein + IgG) or with only the Anti-6× His tag antibody for the MAF1 protein (Ab only). DNA isolated from the immunoprecipitated protein–DNA complex was subjected to qPCR. Deletion of a SINE in the *CDKN1A* template as well as deletion or mutation of the Pol III A-box element in the *CDKN1A* and *GDF15* template depleted MAF1 binding. Binding of MAF1 to *RPPH1* or *ACTB* promoters was not detected. Data shown are the mean ± SD, $n \geq 3$, **$p < 0.01$, ***$p < 0.001$ ($t$-test). (**C**) An in vitro DNA–protein binding assay was performed using a colorimetric assay kit (ab117139). The assayed DNA template 'p21' (DNA template with a Pol III A-box element obtained from *CDKN1A*) was labeled with biotin (a probe). Purified MAF1 protein (His tag) (80R-1955, Fitzgerald) was used for the binding assay. Different competitors (described below) were added to the mixture to demonstrate the specificity of binding of MAF1 at the Pol III promoter element. Competitors: 'self' indicates the same DNA template without the biotin label, 'GDF' indicates the non-labeled DNA template that contained the Pol III promoter element obtained from the *GDF15* promoter, and 'Mut' indicates the Pol III A-box element was mutated in the DNA template. A blank control was performed without the addition of protein, and the degree of enrichment was calculated by subtracting the value of the blank control. MAF1 directly bound to the Pol III promoter element, but the mutant form did not. Data shown are the mean ± SD, $n = 3$, ***$p < 0.001$ ($t$-test). (**D**) In vitro transcription assays were performed on *CDKN1A* and *TAF5* using the HeLaScribe[R] Nuclear Extract in vitro Transcription System (Promega), as indicated in the 'Materials and methods'. Inhibition of Pol II transcription was performed by addition of α-amanitin during in vitro transcription of *CDKN1A* and *TAF5*. The MAF1 protein was pre-incubated with template DNA before addition of nuclear extract to enable binding of MAF1 to the template DNA. (**E**) Different antibodies, as indicated, were pre-incubated with nuclear extract before adding template DNA to perform in vitro transcription to deplete the target protein of interest. For the control, no antibody was added prior to in vitro transcription. In vitro transcription performed on Pol III-transcribed *RPPH1* and Pol II-transcribed *TAF5* served as controls. In vitro transcription performed

*Figure 4. continued on next page*

Figure 4. Continued

on *CDKN1A* and *GDF15* revealed that removal of MAF1 promoted transcription, whereas A-box-deleted *GDF15*, denoted as 'GDF15 (Del)', did not. The degree of enrichment of all performed in vitro transcription was calculated relative to the ratio of signals obtained from the input RNA after subtraction of the negative control (no biotin labeling). All data shown represent the mean ± s.e.m., *n* ≥ 3, *p < 0.05, **p < 0.01, ***p < 0.001 (*t*-test).

repressor of *CDKN1A* transcription. As a control, pre-incubation with MAF1 protein did not affect *TAF5* transcription (*Figure 4D*) because MAF1 did not bind to this DNA template in vivo (*Figure 3I*).

When nuclear extract was pre-incubated with an anti-MAF1 antibody to deplete MAF1 during in vitro transcription, *CDKN1A* transcription was significantly upregulated compared with that of the control, which was pre-incubated with IgG or no antibody (*Figure 4E*). Simultaneous depletion of Pol III and MAF1 by pre-incubation nuclear extract with Pol III and MAF1 antibodies abolished the enhancement of transcription after depletion of MAF1 alone (*Figure 4E*). Specificity of the Pol III antibody was demonstrated by the inhibitory effect of anti-Pol III antibody on in vitro transcription of the Pol III-transcribed *RPPH1* promoter in a pSUPER plasmid. Addition of the anti-MAF1 antibody did not induce *RPPH1* transcription (*Figure 4E*) because binding of MAF1 was not detected in this gene by the in vitro MAF1 binding assay (*Figure 4B*).

These experiments recapitulated the in vivo transcription regulation of *CDKN1A* by MAF1 and Pol III. Taken together with the in vivo and in vitro nuclear run-on expression analyses, these results unambiguously demonstrate that MAF1 can serve as a repressor of the *CDKN1A* promoter, and that recruiting Pol III after MAF1 depletion is crucial for the activation of *CDKN1A* transcription.

## MAF1 knockdown promoted recruitment of positive regulatory factors and induced histone modifications associated with gene activation

The data above indicate that binding of Pol III to the *CDKN1A* promoter after *MAF1* knockdown is crucial for enhanced transcription. This indicates that Pol III may help recruit the regulatory factors necessary for efficient Pol II transcription. To test this hypothesis, we carried out ChIP analysis to examine the Pol III-dependent recruitment of transcription activators after MAF1 removal. ChIP analysis showed that *MAF1* knockdown resulted in significantly enhanced levels of active histone modifications, including H3K4me3, H3K9Ace, and H3K27Ace, in the 5′ regions of *CDKN1A* (*Figure 5A,B*). The enhanced active histone marks after *MAF1* knockdown were abolished under simultaneous knockdown of Pol III and *MAF1* (*Figure 5B*). The histone repression marker, H3K27me3, was detected at the 5′ regions and decreased after *MAF1* knockdown, but the level was restored under simultaneous knockdown of Pol III and *MAF1* (*Figure 5B*).

Because H3K4 methylation is catalyzed by the SET1/MLL family of histone methyltransferases in humans (*Shilatifard, 2012*), we performed knockdown assays to investigate which methyltransferase is responsible for H3K4me3 modification after *MAF1* knockdown. Previously, CFP1 (SET1C-specific subunit) and p300 were shown to act cooperatively to regulate H3K4me3 modification and *CDKN1A* transcription (*Tang et al., 2013*). Indeed, the *MAF1* knockdown-induced transcription of *CDKN1A* was downregulated after *CFP1* (CXXC1) knockdown (*Figure 5—figure supplement 1A*).

p300 and PCAF have been shown to regulate K27 and K9 acetylation, respectively, of the *CDKN1A* promoter (*Love et al., 2012*). Thus, we next analyzed whether CFP1, p300, and PCAF could bind to the *CDKN1A* promoter after *MAF1* knockdown. As expected, the removal of MAF1 by knockdown induced binding of CFP1, p300, and PCAF to the *CDKN1A* promoter (*Figure 5C–E*). Furthermore, simultaneous knockdown of *MAF1* and Pol III abolished the induced binding of these factors along with active histone marks, which indicates that Pol III is required to recruit these factors to the *CDKN1A* promoter for histone modifications (*Figure 5C–E*).

Because TBP is an important factor required in both Pol II (part of TFIIB) and Pol III (part of TFIIIB) transcription (*Zhao et al., 2003*), we next determined whether binding of TBP was enhanced after *MAF1* knockdown. Indeed, enhanced binding of TBP was observed at the *CDKN1A* promoter after *MAF1* knockdown, and the binding was abolished when there was simultaneous knockdown of Pol III and *MAF1* (*Figure 5F*). Simultaneous knockdown of *MAF1* and *TBP* also abolished the enhanced expression of *CDKN1A* by knockdown of only *MAF1* (*Figure 5—figure supplement 1B*), which indicates that TBP is important for *CDKN1A* transcription activation. We also observed enhanced binding of a common subunit of all three RNA polymerases, that is, POLR2E (RPB5), after *MAF1*

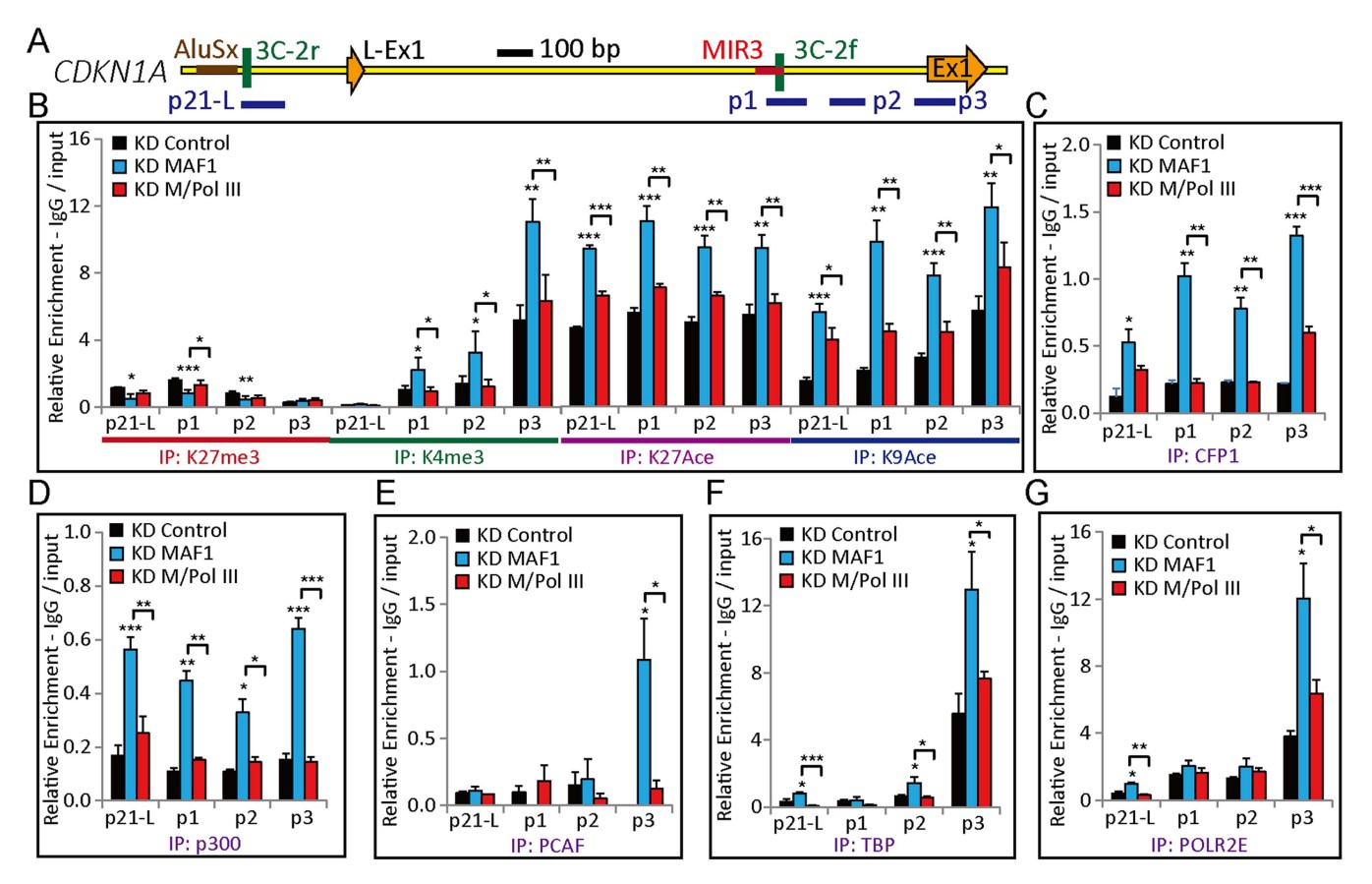

**Figure 5**. *MAF1* knockdown induces Pol II initiation, active histone marks (H3K4me3, H3K9Ace, and H3K27Ace), and binding of CFP1, p300, PCAF, TBP, and POLR2E at the *CDKN1A* promoter. (**A**) Diagram of the *CDKN1A* promoter, including locations of exon 1 (Ex1), SINEs (AluSx and MIR3), and ChIP qPCR amplicons (p21-L, p1, p2, and p3). (**B**) Knockdown coupled with ChIP assays with antibodies for H3K27me3, H3K4me3, H3K27Ace, and H3K9Ace were performed in MCF-7 cells subjected to siRNA knockdown for 72 hr. DNA isolated from immunoprecipitated chromatin was subjected to qPCR and calculated as described in the 'Materials and methods'. Knockdown *MAF1* (KD MAF1) enhanced active histone marks H3K4me3, H3K27Ace, and H3K9Ace, whereas simultaneous knockdown of Pol III and *MAF1* (KD M/Pol III) abolished the enhanced histone marks. ChIP with anti-CFP1 (IP: CFP1) (**C**), anti-p300 (IP: p300) (**D**), anti-PCAF (IP: PCAF), (**E**) anti-TBP (IP: TBP) (**F**), and anti-POLR2E (IP: POLR2E) (**G**) antibodies were performed as described in (**B**). Knockdown *MAF1* (KD MAF1) enhanced binding of CFP1, p300, PCAF, TBP, and POLR2E, whereas simultaneous knockdown of Pol III and *MAF1* (KD M/Pol III) abolished the enhanced binding. All data shown are the mean ± s.e.m., $n \geq 3$, *p < 0.05, **p < 0.01, ***p < 0.001 (*t*-test).

The following figure supplement is available for figure 5:

**Figure supplement 1**. Enhanced gene expression by *MAF1* knockdown is abolished by simultaneous knockdown of *MAF1* with *TBP* or *CFP1*.

knockdown, and this enhanced binding was also abolished when there was simultaneous knockdown of Pol III and *MAF1* (*Figure 5G*), indicating interplay between Pol II and Pol III polymerases.

## Pol III is required for chromatin looping at the *CDKN1A* promoter after *MAF1* knockdown

Because binding of Pol III and Pol II was detected at the 5′ flanking regions of both long and short *CDKN1A* forms in the UCSC Genome Database after *MAF1* knockdown, we performed 3C analysis to investigate whether chromatin looping occurs between these two regions. Chromatin looping was detected between the regions after *MAF1* knockdown, but not in adjacent regions (*Figure 6A,B*). Moreover, simultaneous knockdown of *MAF1* with either Pol III or *BRF1* (a subunit of TFIIIB) disrupted the looping formation (*Figure 6C*). These results demonstrate that Pol III is required for induced chromatin looping after *MAF1* knockdown.

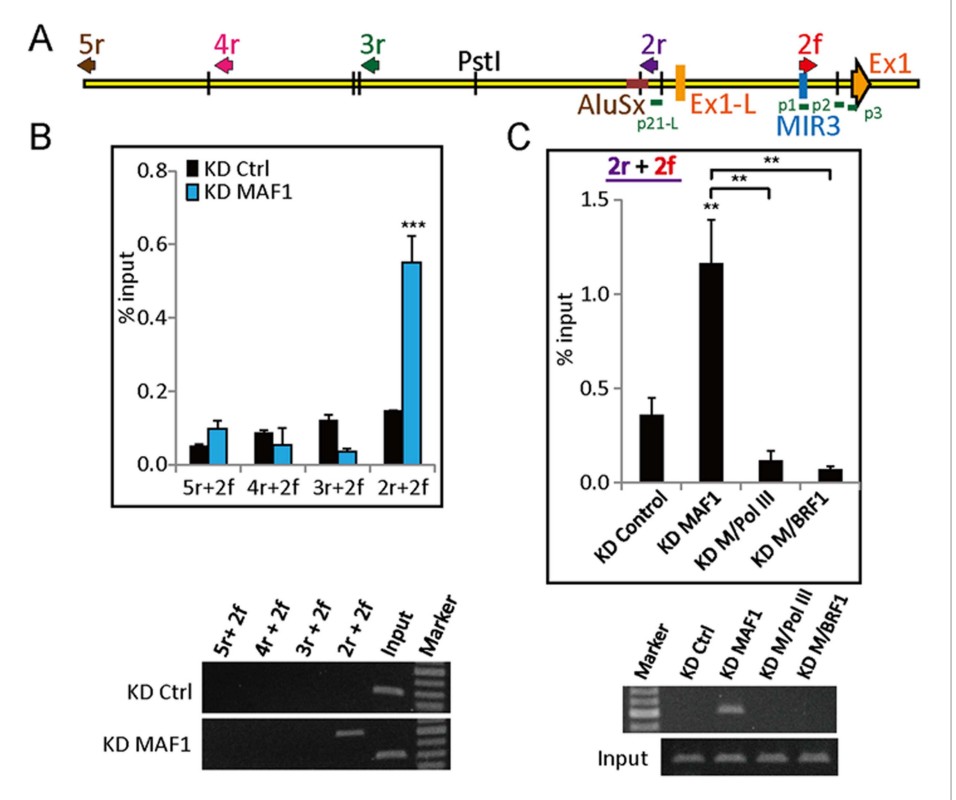

**Figure 6**. Pol III is required for chromatin looping at the *CDKN1A* promoter after *MAF1* knockdown. (**A**) Schematic diagram of *CDKN1A* with the orientation of 3C primers (arrows: 5r, 4r, 3r, 2r, and 2f) and location of exon 1 (long form: L-Ex1; short form: Ex1). (**B**) MCF-7 cells were subjected to siRNA knockdown of *MAF1* (KD MAF1) for 72 hr. 3C assay was performed as indicated in the 'Materials and methods', and DNA was subjected to PCR. Chromatin looping was detected after *MAF1* knockdown from 2r to 2f (top panel) and are shown by a representative gel (bottom panel). (**C**) The induced chromatin looping after *MAF1* knockdown was diminished when either Pol III (KD M/Pol III) or *BRF1* (KD M/BRF1) underwent simultaneous knockdown with *MAF1* (top panel) and are shown by a representative gel (bottom panel). All data shown represent the mean ± s.e.m., $n \geq 3$, *$p < 0.05$, **$p < 0.01$, ***$p < 0.001$ (*t*-test).

## Pol III is required for transcriptional activation and chromatin looping of *GDF15* after *MAF1* knockdown

The above results demonstrate that *MAF1* knockdown can activate *CDKN1A* expression by recruiting Pol III and Pol II along with histone-modifying factors. To demonstrate that this type of mechanism also regulates expression of other Pol II genes, we performed expression analysis of *GDF15*, which is another cell proliferation-related gene that is upregulated after *MAF1* knockdown as found by microarray analysis. As expected, qRT-PCR analysis showed that *GDF15* expression was strongly upregulated after *MAF1* knockdown, and simultaneous knockdown of *MAF1* with Pol III diminished the induced expression (*Figure 1D*). ChIP analysis also indicated binding of MAF1 at the 5′ flanking region of *GDF15* (*Figure 7A,B*).

We also employed an in vitro transcription assay using HeLa cell nuclear extract to demonstrate the importance of the Pol III promoter element in regulation of *GDF15* transcription after *MAF1* knockdown. When nuclear extract was pre-incubated with anti-MAF1 antibody to deplete MAF1 during in vitro transcription, *GDF15* transcription was significantly upregulated compared with the control with pre-incubation with IgG or no antibody (*Figure 4E*). We also noticed an MIR repeat element in the 5′ flanking region of *GDF15*. As in the case of *CDKN1A*, deletion of the Pol III A-box element associated with this repeat also abolished the enhancement of *GDF15* in vitro transcription when anti-MAF1 antibody was added to the extract (*Figure 4E*), indicating that this element mediated

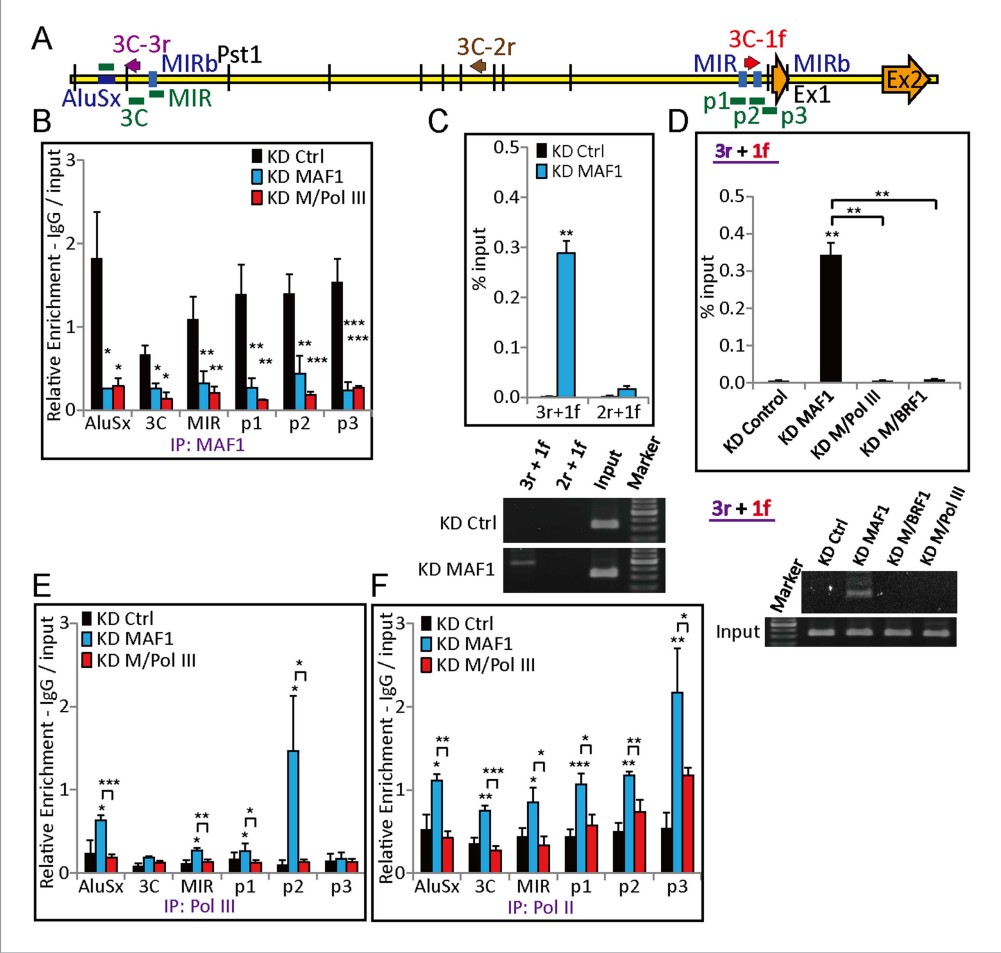

**Figure 7**. Pol III is required for chromatin looping at the *GDF15* promoter after *MAF1* knockdown. (**A**) Schematic diagram of *GDF15* with ChIP–qPCR amplicons (AluSx, 3C, MIR, p1, p2, and p3), the orientation of 3C primers (arrows: 3C-3r, 3C-2r, and 3C-1f), and locations of exons (Ex1 and Ex2). (**B**) ChIP with anti-MAF1 antibody (IP: MAF1) was performed in MCF-7 cells subjected to siRNA knockdown of *MAF1* (KD MAF1) or simultaneous knockdown of *MAF1* and Pol III (KD M/Pol III) for 72 hr. Binding of MAF1 was detected at the *GDF15* promoter, which diminished after *MAF1* knockdown. (**C**) A 3C assay was performed as indicated in the 'Materials and methods', and DNA was subjected to PCR. Chromatin looping was detected after *MAF1* knockdown from 3C-3r to 3C-1f (top panel) and is shown by a representative gel (bottom panel). (**D**) The induced chromatin looping after *MAF1* knockdown (KD MAF1) was diminished when *MAF1* underwent simultaneous knockdown with either Pol III (KD M/Pol III) or *BRF1* (KD M/BRF1) (top panel) and is shown by a representative gel (bottom panel). (**E**) ChIP with anti-Pol III antibody (IP: Pol III) or anti-Pol II antibody (IP: Pol II) was performed in MCF-7 cells subjected to siRNA knockdown. Enhanced binding of Pol III was detected at the *GDF15* promoter after *MAF1* knockdown, which was abolished when there was simultaneous knockdown of Pol III and *MAF1* (KD M/Pol III). (**F**) *MAF1* knockdown indicates enhanced binding of serine 5-phosphorylated Pol II, which was abolished when there was simultaneous knockdown of Pol III and *MAF1*. All data shown represent the mean ± s.e.m., $n \geq 3$, *$p < 0.05$, **$p < 0.01$, ***$p < 0.001$ (*t*-test).

MAF1 binding. Indeed, the in vitro binding assay using purified MAF1 protein indicated that MAF1 did not bind to this deletion mutant but did bind to the wild-type sequence (*Figure 4B*).

A 3C assay was performed to further investigate whether *MAF1* knockdown induces chromatin looping. The analysis indicated that there was chromatin looping between a promoter region and a region that is 12-kb upstream of the *GDF15* promoter after *MAF1* knockdown (*Figure 7C*). Furthermore, the looping was abolished under simultaneous knockdown of *MAF1* with either Pol III or BRF1 (*Figure 7D*). Similar to the *CDKN1A* results, induced binding of Pol III and Pol II to the *GDF15* promoter was observed after *MAF1* knockdown, whereas the binding was diminished after

simultaneous knockdown of Pol III and *MAF1* (*Figure 7E,F*). These results show that transcription of *GDF15*, like *CDKN1A*, is upregulated after *MAF1* knockdown by recruiting Pol III, and Pol III is required for chromatin looping at the *GDF15* promoter.

## Demonstration of MAF1- and Pol III-mediated transcription regulation using a reporter gene assay

To demonstrate the role of the Pol III promoter element in Pol III-mediated activation of the Pol II gene, we analyzed the effect of deletion of the Pol III promoter element using a reporter assay. The promoter regions of *CDKN1A*, *GDF15*, and *TAF5*, as indicated in the 'Materials and methods', were cloned into the reporter plasmid pGL3. Promoter-driven luciferase activities of *CDKN1A* and *GDF15* were upregulated, whereas *TAF5* promoter-driven expression was not affected after *MAF1* knockdown (*Figure 8A*). Simultaneous knockdown of both Pol III and *MAF1* abolished the upregulation caused by knockdown of *MAF1* alone (*Figure 8A*). These results thus recapitulated the results of in vivo endogenous gene analysis.

Because the majority of SINEs are transcribed by the type II internal Pol III promoter that contains an A-box and B-box (*Okada and Ohshima, 1995*), our model indicates that mutation of the Pol III promoter element in the promoter-associated SINE should abolish the enhancement of reporter expression after *MAF1* knockdown. To test this possibility, we chose the *GDF15* promoter for analysis because it was shown (*Ichikawa et al., 2008*) that deletion of the −465 to −429 sequence, which contains the Pol III promoter element, did not affect promoter activity. We mutated the A-box (−447 to −437) in the *GDF15* promoter (−889 to +110) in the reporter and examined the effect of mutation on reporter expression. Under regular cell culture conditions, no significant change in *GDF15* promoter activity was observed in our results when the A-box was mutated or deleted, in consistent with the results of *Ichikawa et al. (2008)*. However, deletion of A-box in the *GDF15* promoter diminished the upregulation of the reporter after *MAF1* knockdown (*Figure 8B*). These results further demonstrate that MAF1 represses *CDKN1A* and *GDF15* promoter activity by binding to the Pol III promoter element. Moreover, recruitment of Pol III after MAF1 depletion is crucial for transcription activation of these genes.

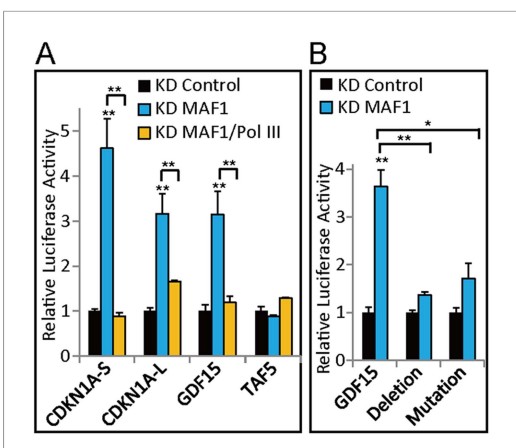

**Figure 8**. Demonstration of MAF1- and Pol III-mediated transcription regulation using a reporter gene assay. (**A**) Promoter regions of Pol II genes were constructed and cloned into pGL3-basic reporter plasmids, as indicated in the 'Materials and methods'. Luciferase reporter assays were performed in MCF-7 cells subjected to siRNA knockdown of *MAF1* or simultaneous knockdown of Pol III and *MAF1*. Results are normalized with β-galactosidase and presented relative to knockdown control cells transfected with pGL3-basic. (**B**) The consensus sequence of the A-box (−447 to −437) in the *GDF15* promoter (−889 to +110) was either deleted or mutated. Reporter assays were performed in MCF-7 cells subjected to siRNA knockdown of *MAF1* (KD MAF1). All data shown represent the mean ± s.e.m., $n \geq 3$, *p < 0.05, **p < 0.01, ***p < 0.001 (*t*-test).

## Discussion

In this research, we showed that MAF1 bound to promoter-associated SINEs associated with type II Pol III promoters and that depletion of MAF1 enhanced transcription activity and chromatin looping by the recruitment of Pol III along with active Pol II and factors associated with these promoters. Both in vivo gene expression and R-looping analysis as well as in vitro transcription using the HeLa nuclear extract and in vitro binding using purified MAF1 protein revealed that MAF1 represses *CDKN1A* and *GDF15* promoter activity by binding to the SINE repeats within their promoters. This result strongly indicates a novel transcription regulatory mechanism whereby MAF1 also acts as a specific repressor of some Pol II genes by binding to promoter-associated SINEs. Binding specificity was demonstrated by an in vitro DNA binding assay with purified MAF1 to wild-type *CDKN1A* and *GDF15* promoters and the lack of binding to

promoters with mutations in the SINE. To the best of our knowledge, this is the first time that MAF1 has been shown to bind to specific DNA sequences.

We also demonstrated that recruiting Pol III to SINEs in the 5′ flanking region is required to promote Pol II gene transcription, epigenetic modifications, and chromatin looping after *MAF1* knockdown. Recruitment of Pol III, positive regulatory factors, and common transcription factors (TBP and POLR2E) of Pol II and Pol III demonstrate a novel mechanism of activating Pol II genes through Pol III-mediated activation mechanism. Gene expression induced by chromatin remodeling through SINEs has also been described in neuronal genes that undergo acetylation of distal promoter SINEs by p300 to translocate genes to transcription factories (*Crepaldi et al., 2013*). In our result, *MAF1* knockdown promoted recruitment of p300, which has been shown to promote acetylation of histone K27 and active transcription by Pol II. However, how extensively this mechanism regulates genes is currently unknown.

Histone H3K4me has been suggested to serve as a hallmark of enhancer (*Herz et al., 2012*). Examination of ENCODE database revealed that this epigenetic mark was presented in both 5′ flanking regions of *CDKN1A*. Indeed the identified p53-binding site located between 1.4 kb and 2.3 kb upstream of *CDKN1A* has been identified as the enhancer region (*Melo et al., 2013*; *Leveille et al., 2015*). However, the SINE with the MAF1-binding site located at 2.65 kb upstream of short promoter exerted no enhancer activity in the luciferase assay prior to or after *MAF1* knockdown (data not shown). The chromatin looping we observed for *CDKN1A* and *GDF15* after *MAF1* knockdown may be mediated through the proximity and interaction between the sets of transcriptional factors recruited in the two 5′ flanking regions after chromatin remodeling as proposed by *Crepaldi et al. (2013)*.

Because the background expression of Pol II genes slightly decreased when Pol III was knockdown, Pol III may be able to regulate the expression of some minor alleles without MAF1 being bound to the promoter. However, it is difficult to unequivocally validate this possibility without specific technology that can efficiently separate different alleles in cells.

Human genome analysis indicated that 71% of genes contained SINEs in their promoter regions. Microarray analysis showed that 124 genes were upregulated after *MAF1* knockdown. Of these, 76% contained SINEs within the promoter region, which indicates the regulation potential of these genes by MAF1 and Pol III. However, microarray analysis of steady-state mRNA level alone is not sufficient to show whether these genes are directly regulated by MAF1 at the transcriptional level or as the result of downstream secondary effects. Further analysis using in vitro transcription, reporter gene analysis, and nuclear run-on would be required to unambiguously establish how general Pol II genes are regulated by MAF1 and Pol III. In vitro transcription using HeLa cell extracts with depletion or addition of specific transcription regulators could provide very strong support of the involvement of specific factors in transcription regulation.

Close proximity of Pol III genes to Pol II genes has been observed genome-wide (*Oler et al., 2010*). Active Pol III-transcribed genes and non-coding RNAs often associate with Pol II transcription start sites. Pol II and its associated epigenetic marks are also present at active Pol III-transcribed genes (*Barski et al., 2010*; *Raha et al., 2010*; *Canella et al., 2012*). This shows that there is common epigenetic regulation between these two types of transcription units, and the polymerases may work with one another to regulate gene expression. Indeed, cross-talk between Pol III and Pol II transcription factors, such as TFIIS in Pol III transcription, has also been reported in yeast and mice (*Ghavi-Helm et al., 2008*; *Carriere et al., 2011*). The core Pol III transcription factor TFIIIC can also directly regulate transcription from a Pol II promoter (*Kleinschmidt et al., 2011*). Binding of TFIIIC to SINE promoters has been shown to mediate the relocation and transcription of neuronal genes (*Crepaldi et al., 2013*). *RPPH1*, to which BRF2, Pol III, GTF2B, and Pol II bind, can be transcribed by either Pol II or Pol III (*James Faresse et al., 2012*). Our results indicate that Pol III and Pol II association may have functional relevance for genome functional organization, because simultaneous knockdown of both Pol III and *MAF1* diminished the induced active transcription caused by knockdown of only *MAF1*. Furthermore, enhanced binding of TBP to TFIIIB and TFIIB can lead to formation of Pol III and Pol II complexes to initiate transcription (*Zhao et al., 2003*). We propose that this type of SINE-associated-Pol II promoter architecture may introduce an additional layer of control in gene expression.

Recently, MAF1 was shown to be a negative regulator of transcription of all three polymerases, Pol I, Pol II, and Pol III, through mediating *TBP* expression (*Johnson et al., 2007*). Johnson et al. showed

that MAF1 binds to the Elk-1-binding site of the *TBP* promoter to prevent the binding of Elk-1. Indeed, there is a SINE with an A-box ($-10$ to $+1$) and B-box ($-105$ to $-95$) that encompass the Elk-1-binding site of the *TBP* promoter. In our analysis, *MAF1* knockdown only slightly increased *TBP* expression (1.6-fold) compared with the results reported by Johnson et al. (twofolds). This may be due to the already high expression of *TBP* in cell lines, and we did not detect binding of MAF1 to this active promoter.

Based on our results, we propose the following mechanism of control of Pol II gene transcription by MAF1 and Pol III: Before removal of MAF1 from SINEs, Pol II is in a transcriptionally engaged but paused state, where TBP/TFIIB is pre-assembled and remains at the promoter (*Guenther et al., 2007*; *Kwak et al., 2013*; *Venters and Pugh, 2013*). Removal of *MAF1* by knockdown then leads to recruitment of TFIIIB through enhanced binding of TBP; the shared surface of TBP then directs both Pol II and Pol III binding through association with TFIIB and TFIIIB, respectively. Further recruitment of active regulatory factors would then induce transcription by Pol II and Pol III. This model is consistent with those proposed by previous studies, which were based on a component of TBP-associated complexes, p300, interacting with SET1C-coupled histone modifications to activate *CDKN1A* transcription (*Abraham et al., 1993*; *Tang et al., 2013*). Moreover, our model is also supported by a previous study on the relocation of inducible neuronal genes to transcription factors that involve acetylation of distal promoter SINEs by p300 (*Crepaldi et al., 2013*).

# Materials and methods

## Cell culture

MCF-7, MCF-10A, and MDA-MB-231 cell lines were originally obtained from ATCC (Manassas, VA), and cultured in RPMI, HuMEC and DMEM medium (Invitrogen; Waltham, MA), respectively. HCT-116$^{p53+/+}$ (wild-type) and HCT116$^{p53-/-}$ (p53-null) cell lines were originated from Bert Vogelstein (John Hopkins University) and cultured in McCoy's 5A medium (*Bunz et al., 1998*). Each medium was supplemented with 10% of fetal bovine serum and incubated in a humidified 37°C incubator with 5% $CO_2$.

## RNAi knockdown assay

Knockdown assay was performed using siRNA obtained from MISSION RNA (Sigma-Aldrich; St. Louis, MO). Inhibition of expression of *MAF1* ([#1] SASI_Hs01_00135954, [#2] SASI_Hs01_00135956 and [#3] SASI_Hs01_00135958), Pol III (POLR3A) ([#1] SASI_Hs01_00046568, [#2] SASI_Hs01_00046571 and [#3] SASI_Hs01_00046572), *BRF1* (SASI_Hs01_00131187), *CFP1* (SASI_Hs02_00322879), and *TBP* (SASI_Hs01_00122768) was achieved by transfection with Lipofectamine RNAiMax (Invitrogen) according to the manufacturer's protocol for 72 hr. MISSION siRNA Universal Negative Control (Sigma) was used as knockdown control. Cells were transfected in serum-free medium. After 8 hr, the siRNA containing medium was replaced with complete medium.

## Immunoblotting

Cells were lysed at 4°C in RIPA lysis buffer (50 mm Tris-HCl, pH 7.2, 150 mm NaCl, 5 mm EDTA, 1% [wt/vol] NP-40, 1% [wt/vol] SDS and protease and phosphatase inhibitor mixtures [Roche Applied Science; Penzberg, Germany]). The lysates were cleared by centrifugation (15,000×*g* for 15 min), resolved on a 10% SDS-polyacrylamide gel, and transferred onto a nitrocellulose membrane. The antibody dilutions used were rabbit anti-POLR3A (1:1000; ab96328, Abcam; Cambridge, England), rabbit anti-MAF1 (1:1000; GTX106776, Acris; Herford, Germany), rabbit anti-CDKN1A (1:1000; ab18209, Abcam), and mouse anti-tubulin (1:10,000; ab7291, Abcam).

## RNA extraction

Cells were grown to 85% confluence in 6 cm tissue culture dish. Each 6 cm dish was washed with 1× phosphate buffered saline (PBS) for three times. Total RNA was extracted using TRIreagent (Invitrogen) protocol. The integrity of the RNA extract was checked by 1.2% (wt/vol) agarose gel electrophoresis and the concentration of RNA was estimated by ultraviolet spectrophotometry.

## Microarray

Affymetrix microarray was performed using Human U133 plus 2.0 (Affymetrix; Santa Clara, CA). Details of the methods for RNA quality, sample labeling, hybridization, and expression analysis were

according to the manual of Affymetrix Microarray Kit. All Affymetrix data are MIAME compliant and that the raw data have been deposited in a MIAME compliant database, GEO. The microarray data were deposited at the NCBI GEO website (GEO accession number GSE42239).

## Quantitative RT-PCR

Reverse transcription was performed by using superScript III RNase H- Reverse Transcriptase (Invitrogen) and random hexamer according to the manufacturer's protocol. Quantitative PCR was performed using KAPA SYBR FAST (KK4603, KAPA Biosystems; Wilmington, MA) on ABI StepOnePlus Real-Time PCR System (Invitrogen). All reactions were performed in triplicate with KAPA SYBR FAST plus 10 μM of both the forward and reverse primer according to the manufacturer's recommended thermo cycling conditions, and then subjected to melting curve analysis. The calculated quantity of the target gene for each sample was divided by the average sample quantity of the housekeeping genes, glyceraldehydes-3-phosphate dehydrogenase (GAPDH) or 18S to obtain the relative gene expression.

## Flow cytometry analysis

MCF-7 knockdown cells were collected by trypsinization and washed twice with ice-cold PBS. The cells were resuspended in 0.3 ml of PBS and fixed by slowly adding 3 ml of 70% cold ethanol. Cells were fixed at −20°C for 1 hr. The fixed cells were washed with ice-cold PBS and rehydrated for 15 min. After centrifuging at 200×$g$ for 5 min, cells were resuspended in 0.1 mg/ml of propidium iodide and 0.6% of Triton X-100 in 500 μl of PBS. Then add 500 μl of 2 mg/ml of RNase A and incubate in the dark for 45 min. Data were collected using a FACScan flow cytometry system (BD; Franklin lakes, NJ).

## Nuclear run-on assay

Nuclear run-on reactions were performed by supplying biotin-16-UTP to nuclei, and labeled transcripts were bound to streptavidin-coated magnetic beads as described by *Patrone et al. (2000)* with minor modifications. Nuclei were prepared from MCF-7 cells by resuspension in Nonidet P-40 lysis buffer (10 mM HEPES, pH 7.3, 10 mM NaCl, 3 mM $MgCl_2$, 150 mM sucrose, and 0.5% Nonidet P-40). Nuclei were isolated, and the pellets were resuspended in 1 ml of glycerol buffer (50 mM Tris-Cl, pH 8.3, 40% glycerol, 5 mM $MgCl_2$, and 0.1 mM EDTA). 1 ml of transcription buffer (20 mM Tris-Cl, pH 8.0, 200 mM KCl, 5 mM $MgCl_2$, 4 mM dithiothreitol, 4 mM each of ATP, GTP, and CTP, 200 mM sucrose, and 20% glycerol) was added in the nuclei along with 10 μl of biotin-16-UTP or UTP for run-on reaction or negative control, respectively (Roche). After incubation at 29°C for 30 min, the reaction was terminated by the addition of 12 μl of 250 mM $CaCl_2$, and 12 μl of RNase-free DNase I and incubated at 29°C for 10 min. To purify RNA, a TRIreagent extraction, phenol-chloroform extraction, and isopropanol (Sigma) precipitation were then performed. A small aliquot (5 μl from a total of 50 μl) was saved as input control. Dynabeads M-280 streptavidin (Invitrogen) were mixed with an equal volume of the isolated RNA samples for 20 min at 42°C for 20 min and 2 hr at room temperature. After washing with 15% formamide and 2× SSC, the beads were resuspended in 45 μl of nuclease-free water. Reverse transcription was performed by using superScript III RNase H—Reverse Transcriptase (Invitrogen). Total cDNA was then synthesized by means of random hexamer primed reverse transcription of captured molecules. The gel pictures were quantified with ImageJ (provided by NIH: http://imagej.nih.gov/ij/). The purified run-on products where normalized with internal control (GAPDH) to obtain the relative transcription levels for each gene.

## Detection of R loops using non-denaturing bisulfite treatment

Knockdown assay was performed using siRNA obtained from MISSION RNA (Sigma). Inhibition of expression of Pol III (SASI_Hs01_00046568) and *MAF1* (SASI_Hs01_00135954) was achieved by transfection with Lipofectamine RNAiMax (Invitrogen) according to the manufacturer's protocol for 72 hr. DNA purification and single-stranded R loop foot-printing were carried out as previously described with slight modifications (*Yu et al., 2003*). 500 ng of purified genomic DNA was bisulfite converted by adding CT Conversion Reagent from the EZ DNA Methylation-Gold Kit (Zymo Research; Irvine, CA) at 37°C for 16 hr in the dark. PCR amplified region for cloning is shown in *Figure 2A,F* as foot-printing region. The PCR product was gel eluted and ligated to sequencing vector yT&A (Sigma). Approximately, 20 individual clones were sequenced for all PCR products, and the sequencing data were analyzed and aligned to *CDKN1A* or *ACTB* genomic sequence. The sequence of the beginning

and end of each clone is trimmed due to low quality of sequencing. A background conversion (approximately 5% of cytosine) may be seen possibly due to DNA breathing during the prolonged incubation at 37°C in our data and others (*Yu et al., 2003*). Approximately, 1–2 clones showed both cytosine to thymine and guanine to adenine conversions, which is known as 'mosaic molecules' (*Yu et al., 2003*).

## ChIP and qPCR

ChIP assay was performed according to the manufacturer's protocol (Upstate Biotechnology, Inc.; Lake Placid, NY) with slight modifications. Human MCF-7 cells were fixed with 1% of formaldehyde at room temperature for 10 min. The cells were lysed and the chromatin was sonicated to 200–500 bp fragments by Bioruptor sonicator (cycle condition of 25 s on and 25 s off in a total of 25 min at highest output). Chromatin was immunoprecipitated by using Pol III (ab96328, Abcam), Pol II (ab5131, Abcam), MAF1 (GTX106776, Acris), H3K4me3 (04–745, Millipore; Billerica, MA), H3K27me3 (ABE44, Millipore), TBP (ab28175, Abcam), H3K9Ace (06–942, Millipore), H3K27Ace (07–360, Millipore), CFP1 (ABE211, Millipore), p300 (05–257, Millipore), POLR2E (ab180151, Abcam) BRF1 (ab74221, Abcam)or IgG (ab46540, Abcam) antibody, with 10 µg/ml of BSA and 50 µl of Dynabeads Protein A and G (Invitrogen) for overnight at 4°C. The beads were washed once with each washing buffer, including low salt immune complex wash buffer, high salt immune complex wash buffer, and LiCl immune complex wash buffer, and twice with 1× TE buffer. Precipitates were eluted with 1% of SDS and 100 mM of NaHCO$_3$. Proteinase K was added to the samples, and rotated at 65°C for 2 hr followed by 95°C for 10 min and cooled down to room temperature. RNase A was added and samples were incubated at 37°C for 1 hr. After genomic DNA extraction, qPCR was performed. The degree of enrichment is calculated relative to the ratio of signals obtained in the input DNA fraction subtracting IgG-immunoprecipitated DNA.

## 3C assay

3C assay was performed according to (*Dekker et al., 2002*) with some modifications. MCF-7 cells were fixed in 2% formaldehyde for 10 min at room temperature and quench with 0.125 M glycine. After centrifugation for 15 min at 3500 rpm, the cells were suspended in lysis buffer (10 mM Tris-HCl pH 8.0, 10 mM NaCl, 0.2% Nonidet P-40 and 1:500 Complete protease inhibitor cocktail; Roche) for 90 min on ice. Next, the nuclei were pelleted by centrifugation for 15 min at 2500 rpm, resuspended in 500 µl of 1× NEB buffer 4 plus 0.3% SDS and incubated at 37°C for 1 hr. After the addition of Triton-X to a final concentration of 1.8% to sequester the SDS, the mixture was incubated at 37°C for 1 hr, which was followed by the addition of 800 U of PstI and incubation at 37°C overnight to digest the chromatin. The reaction was terminated by adding SDS to a final volume of 1.6% and then the solution heated to 65°C for 20 min. Ligation of DNA in situ was carried out using 0.5–2.0 ng/µl of chromatin in 800 µl of ligation buffer (NEB; Ipswich, MA) plus 1% Triton-X and 30 Weiss Units of T4 ligase (NEB) for 4 hr at 16°C. After reversing of the crosslinks with proteinase K digestion at 65°C overnight, the DNA was purified by phenol-chloroform extraction and ethanol precipitation. The ligation products were detected by PCR using primers located near Pst1 cutting sites. The PCR products were purified from an agarose gel, cloned and sequenced.

## In vitro DNA binding assay coupled with immunoprecipitation and qPCR

*CDKN1A* (with or without MIR3), *ACTB*, *GDF15* (including deleted or mutated A-box) and *RPPH1* template DNA was obtained by PCR followed by gel elution (Qiagen; Venlo, Netherlands) according to the manufacturer's protocol. The deletion of MIR3 was performed as described by PCR-mediated deletion and checked by sequencing (*Lee et al., 2004*). The purified DNA was further used for in vitro DNA binding reactions as described previously with slight modifications (*Britten, 1996*; *Toth and Biggin, 2000*). The in vitro protein-DNA binding assay coupled with immunoprecipitation was performed as following: 20 ng of DNA template, 400 ng of MAF1 protein (His tag) (80R-1955, Cantor Fitzgerald; New York, NY), 400 ng of Anti-6× His tag antibody (ab18184, Abcam), protease inhibitor (539134, Calbiochem; La Jolla, CA), and 200 ng of BSA was added into 50 µl of binding buffer (20 mM HEPES [pH7.6], 150 mM NaCl, 0.25 mM EDTA, 10% glycerol, 0.2% NP40, and 1 mM DTT). A negative control was performed by substituting IgG antibody for Anti-6× His tag antibody (Protein + IgG) or with only the Anti-6× His tag antibody for the MAF1 protein (Ab only). The mixture was rotated at 4°C

for 10 min and on ice for 30 min. 10 µl of Dynabeads Protein G (Invitrogen) was added to the mixture and rotated at 4°C for 10 min and on ice for 30 min. The immunoprecipitated DNA-protein complexes were then washed twice with washing buffer (20 mM Tris [pH 7.5], 0.25 mM EDTA, 10% glycerol, and 0.2% NP40) and once with TE buffer by each rotating at 4°C for 5 min. Elution was performed with 1% of SDS and 0.1 M of NAHCO$_3$. Input DNA was prepared as 1 ng of template DNA (5% of 20 ng). Proteinase K was added to the samples, and rotated at 65°C for 2 hr followed by 95°C for 10 min and cooled down to room temperature. DNA isolated from immunoprecipitated protein-DNA complex was subjected to qPCR. The degree of enrichment is calculated relative to the ratio of signals obtained in the input DNA fraction.

## In vitro DNA-Protein binding colorimetric assay

Biotin-labeled (labeled at 5′) and non-labeled CDKN1A (5′-AATCAACAACTTTGTATACTTAAGTT CAGTGGACCTCAATTTCCTCATCTGTGAAATAAA-3′) as well as mutated A-box template DNA (5′-AATCAACAACTTTGTATA CTTCCCATCCCAAAACCTCAATTTCCTCATCTGTGAAATAAA-3′) was obtained by oligo synthesis from Genomics (Houston, TX). The oligos were annealed and used for in vitro DNA-protein binding assay by the DNA-Protein Binding Assay Kit (Colorimetric) provided by Abcam (ab117139). The assay was performed according to manufacturer's protocol. In brief, 40 ng of biotin-labeled DNA template and 500 ng of MAF1 protein (His tag) (80R-1955, Fitzgerald) was used for the binding assay. For competition assay, 200 ng of competitor DNA was added to the mixture. Anti-6× His tag antibody (ab18184, Abcam) and Goat anti-Mouse IgG2b heavy chain (HRP) antibody (ab97250, Abcam) were prepared and added according to manufacturer's protocol. Blank control was performed without the addition of protein as specified by the kit and the degree of enrichment is calculated by subtracting with blank control.

## Luciferase assay

The upstream promoter regions of CDKN1A (−864 to +41 of NM_001220778 for short form and −1249 to +92 of NM_001220777 for long form), GDF15 (−889 to +110), and TAF5 (−998 to +157) genes were cloned into the pGL3-basic Reporter Vector (Promega; Fitchburg, WI). Knockdown assay was performed as mentioned above for 24 hr before MCF-7 cells were transfected with the plasmids using Lipofectamine LTX (Invitrogen), along with a plasmid expressing β-galactosidase for normalization. The plasmids were transfected for 48 hr, and the cells were lysed and luciferase assay was conducted using the Luciferase Assay System (Promega) using a fluorimetric plate reader.

## In vitro Transcription System

In vitro transcription was performed by using HeLaScribe$^R$ Nuclear Extract in vitro Transcription System (Promega Cat. #E3110) according to the manufacturer's protocol with slight modifications. Template DNA was prepared by linearizing the constructed promoter region of CDKN1A, GDF15, and TAF5 as used in Luciferase assay, as well as RPPH1 promoter as used in in vitro MAF1 binding assay. In vitro transcription was performed by incubation with linear form of constructed promoter region with nuclear extract, transcription buffer, magnesium ion, GTP, CTP, ATP, biotin-16-UTP, RNase inhibitor, and 30 µg of yeast tRNA. Negative control was performed by incubation with non-biotin labeled NTPs. 0.2 µg of α-amanitin was added during in vitro transcription for inhibition of Pol II transcription. 3 µg of MAF1 protein (His tag) (80R-1955, Fitzgerald) used in in vitro MAF1 binding assay was pre-incubated with template DNA before adding nuclear extract to enable binding of MAF1 to the template DNA. Anti-Pol III (ab96328, Abcam), anti-Pol II (ab5131, Abcam), anti-MAF1 (GTX106776, Acris), or anti-IgG (ab46540, Abcam) antibody was pre-incubated with nuclear extract for 15 min to deplete the target protein of interest. After incubation at 30°C for 1 hr, the reaction was terminated by the addition of 175 µl of HeLa Extract Stop Solution (Promega Cat. #E3110). A TRIreagent extraction, phenol-chloroform extraction, and isopropanol (Sigma) precipitation were then performed to purify RNA. A small aliquot (2 µl from a total of 22 µl) was saved as 'total nuclear RNA' for each condition. The biotinylated RNA was isolated using streptavidin-coated magnetic beads as described in Run-on assay. Reverse transcription was performed by using superScript III RNase H- Reverse Transcriptase (Invitrogen). Total cDNA was then synthesized by means of random hexamer primed reverse transcription of captured molecules.

## Acknowledgements

This work is supported by a grant from the Ministry of Education, Aim for the Top University Plan, and by a grant from High-throughput Genome Analysis Core Facility (NSC 102-2319-B-010-001). The authors acknowledge the High-throughput Genome Analysis Core Facility of National Core Facility Program for Biotechnology, Taiwan (NSC 102-2319-B-010-001), for sequencing. The authors declare no competing financial interests.

## Additional information

### Funding

| Funder | Grant reference | Author |
|---|---|---|
| Ministry of Education of the People's Republic of China | Aim for the Top University Plan | Ming-Ta Hsu |
| National Research Program for Biopharmaceuticals (NRPB) | High-throughput Genome Analysis Core Facility (NSC-102-2319-B-010-001) | Ming-Ta Hsu |

The funders had no role in study design, data collection and interpretation, or the decision to submit the work for publication.

### Author contributions

Y-LL, Performed most of the experiments, Conception and design, Acquisition of data, Analysis and interpretation of data, Drafting or revising the article; Y-CL, Prepared samples for Microarray analysis, Contributed unpublished essential data or reagents; C-HS, Performed 3C assays, Acquisition of data; C-HC, Performed reporter assays, Acquisition of data; I-HL, Performed R-loop analysis, Analysis and interpretation of data; M-TH, Drafting or revising the article

### Author ORCIDs

Yu-Ling Lee, http://orcid.org/0000-0003-1897-9918
I-Hsuan Lin, http://orcid.org/0000-0002-6207-1299

## Additional files

### Major dataset

The following dataset was generated:

| Author(s) | Year | Dataset title | Dataset ID and/or URL | Database, license, and accessibility information |
|---|---|---|---|---|
| Lee Y, Li Y | 2015 | MAF1 Represses CDKN1A through a Pol III-Dependent Mechanism | http://www.ncbi.nlm.nih.gov/geo/query/acc.cgi?acc=GSE42239 | Publicly available at the NCBI Gene Expression Omnibus (Accession no: GSE42239). |

Standard used to collect data: All Affymetrix data are MIAME compliant and that the raw data have been deposited in a MIAME compliant database, GEO. The microarray data were deposited at the NCBI GEO website (GEO accession number GSE42239).

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
