## [Decision Letter]

Thank you for sending your work entitled “MAF1-regulated-Pol III-mediated activation of *CDKN1A* through promoter looping” for consideration at *eLife*. Your article has been evaluated by James Manley (Senior editor) and four reviewers, one of whom is a member of our Board of Reviewing Editors.

The Reviewing editor and the other reviewers discussed their comments before we reached this decision, and the Reviewing editor has assembled the following comments to help you prepare a revised submission.

The reviewers found the findings to be potentially exciting as they extend and expand thinking about how protein-coding genes are transcribed, in particular a novel role for RNA polymerase III in this process. Overall, this report presents some exciting new findings that are very timely in view of (i) extensive ChIPseq data in the literature demonstrating the localization of components of the Pol III transcription machinery adjacent to Pol II-transcribed genes and (ii) the widespread localization of SINE elements (of unknown function) in the human genome and near promoters. However, the reviewers found several issues that must be addressed in the revised manuscript before it could be recommended for publication:

1) The possible role of p53. The two genes examined throughout the manuscript, *CDKN1A* and *GDF15*, are the two most potently transcribed p53 target genes (Allen et al., *eLife* 2014). However, the authors do not examine the effects that p53 may play in their system. It is entirely possible that most of the effects described upon *MAF1* knockdown are driven indirectly by activation of p53 downstream of a cellular stress response caused by deregulated RNAPIII function. The authors should check for p53 induction upon *MAF1* knockdown. Key experiments should be repeated in an isogenic system -/+ p53. This could be done with stable p53 knockdown MCF7 cells (Agami et al.) or HCT116 p53 -/+ cells (Vogelstein et al.).

2) Promoter definition. The authors indicate that ‘Examination of *CDKN1A* gene shows that there are two promoter regions (hereafter referred as long and short form)’. This description contrasts with many earlier reports that clearly show a major promoter for *CDKN1A* with minor alternative start sites. What exactly are the long and short forms? What NM genebank reference numbers do they correspond to? Where are they located relative to the well-characterized p53 enhancers? Importantly, the authors show equivalent Ser5-phospho-RNAPII binding to both promoters, but high H3K4me3 and TBP binding are found only at the downstream promoter encoding the ‘short form’, indicating that this promoter is the functional promoter. Confusingly, although the authors show equivalent binding of Ser5-phospho-RNAPII to both promoters, RBP5, a subunit common to all three polymerases is found preferentially at the downstream promoter. How much RNA is being produced from the ‘long’ and ‘short’ promoters? Some form of quantitative assay should be done to define the relative strength and RNA output of these two ‘promoters’.

3) Massive misinterpretation of ‘chromatin looping’ data. The authors repeatedly state throughout the manuscript (including the Title) a cause-effect relationship between chromatin looping and gene activation. They state in a subheading of the results that ‘*MAF1* knockdown enhanced *CDKN1A* gene expression from both long and short form promoters through Pol III-mediated promoter looping’. In the next subheading they repeat: ‘*MAF1* knockdown enhanced *GDF15* gene expression through PolIII-mediated promoter looping’. This is a massive misinterpretation of data, because there is no experiment showing a cause-effect relationship between looping and gene activity, only a correlation. The authors state that ‘Moreover, simultaneous knockdown of *MAF1* with either Pol III or *BRF1* (subunit of TFIIIB) disrupted the looping formation (Figure 5). These observations suggest chromatin looping between the long and short form promoter regions is important for the activation of *CDKN1A* transcription from both promoters’, and ‘The above results demonstrate that *MAF1* knockdown can activate *CDKN1A* expression through promoter looping induced by recruitment of Pol III and Pol II along with histone-modifying factors’. Later on, they state ‘These results further demonstrate unambiguously that *MAF1* protein represses the promoter activity of *CDKN1A* and *GDF15*, and that recruitment of Pol III after *MAF1* depletion is crucial for the activation of transcription of these genes through promoter looping’. The looping observed may as well be a consequence, rather than a driver, of gene activation, or just as well a totally inconsequential event. The manuscript should be rewritten to remove all indication of cause-effect relationship between looping and gene activation.

4) The authors rely on the use of a single cell line, MCF7. As these are breast cancer cells that have sustained multiple genetic alterations, it is unclear whether the findings will hold in normal cells or even in other tumor cell lines. This needs to be addressed as the results with *MAF1* may represent a peculiar result of the specific genetic make-up of this single, genetically complex cell type.

5) The bulk of the findings rely in the use of RNAi approaches but the authors use only a single targeting sequence for the knockdown of each particular target. Multiple targeting sequences need to be used and the extent of downregulation at both the protein and RNA levels needs to be shown to reduce the possibility of off-target effects and to further validate the conclusions.

6) As a general comment both on the cellular knockdown data and the in vitro immuno-depletion data, the authors do not generally indicate the extent of depletion of the factors of interest, and this may well account for some of the modest, albeit significant, effects. Thus it would help to have data/comments on the depletion efficiencies.

7) In the nuclear run-on assay of Figure 1, the effect of the *MAF1* knockdown on *CDKN1A* expression appears relatively modest (especially in relation to the clear large effect in the quantitative RT-PCR analysis of Figure 1). Could the authors provide quantitation of the data for *CDKN1A* (and for *GDF15*, where the up-regulation is larger) and perhaps improved data?

8) The authors state that knocking down Pol III alone did not affect *CDKN1A* expression. However, a close examination the data in Figure 1 suggest that expression of *CDKN1A* is actually reduced about 30% (and that of *GDF15* more than 50%). This fits the model that Pol III can positively regulate the expression of *CDKN1A* and *GDF15*, but is potentially inconsistent with the authors' failure to see any Pol III occupancy prior to *MAF1* knockdown. Is this just due to the lower sensitivity and higher background in the ChIP analysis?

9) In Figure 3, the *MAF1* ChIP signal is only 3-fold above background. Because this data provides important evidence of direct regulation, and even though it is supported by the in vitro demonstration of direct *MAF1* binding to the SINE element, the authors might consider ways to modify the ChIP protocol to increase the signal/background ratio.

10) While the in vitro data for the direct *MAF1* binding to the p21 SINE element (Figure 4) seems convincing, especially with the SINE deletion control, it would be very interesting to have more detailed data on the interaction site in light of the novelty of this observation. Have the authors any further information from DNase footprinting, gel shifts with point mutations in the Pol III promoter elements etc. on this point, especially in relation to their suggestion in the Discussion that “*MAF1* and Pol III/TFIIIB may recognize similar DNA elements”? (It should also be noted that the promoter elements of Pol III-transcribed genes are directly recognized by TFIIIC and not by Pol III/IIIB.)

11) In relation to the interesting (and potentially provocative) new *MAF1* mechanism involving direct binding to promoter elements, the reader is advised (in the Introduction) of prior reports indicating *MAF1* binding to *BRF1* and/or Pol III as well as Pol III/*MAF1* co-occupancy on target genes, and thus anticipates that *MAF1* may be repressing Pol III function by a related mechanism on the target genes analyzed here. This, of course, appears not to be the mechanism from the ChIP analyses that follow the functional studies, but I think the rationale and results could be presented with more clarity and with clear indications that, perhaps surprisingly, they do not fit the original models but suggest an alternate mechanism that is pursued (by the direct binding studies).

[Editors’ note: further concerns were raised before acceptance.]

Thank you for resubmitting your article entitled “MAF1-Regulated-Pol III-Mediated Activation of *CDKN1A* and Chromatin Looping” for further consideration by *eLife*. Your revised submission has been evaluated by the original Reviewing Editor and Senior Editor.

Despite a commendable effort by the authors to try to address the concerns raised by the reviewers, the revised manuscript fails to meet the high standards of scholarship expected for publication in *eLife*. The authors have performed additional experiments in response to the reviewers suggestions, mostly to include missing controls and replicates, but the manuscript still fails to produce a clear picture of how exactly *MAF1* and POLIII regulate protein-coding genes such as *CDKN1A* and *GDF15*.

The major issue is that, in response to a request for clarification from the reviewers, it is now evident that the genomic region that the authors have called the ‘p21 long promoter’ (p21L) is indeed the enhancer regulatory region of this gene, known to be bound by p53 and many other transcription factors. This region, located 2.25kb upstream of the canonical p21 promoter, produces enhancer-derived RNAs (eRNAs) and may engage with the promoter through a chromatin loop. In view of this, the entire dataset must be reinterpreted as *MAF1* and POLIII likely affecting the activity of the enhancer region at the *CDKN1A* locus, and the manuscript should be rewritten in light of this fact.

Finally, the revised manuscript failed to correct some points raised by reviewers. As an example of a major oversight, the authors tested for ‘p53 activation’ by Q-RT-PCR, when it is well established that p53 activation upon cellular stress occurs at the protein level, and thus should have been tested by Western blot.

---

## [Author Response]

*1) The possible role of p53. The two genes examined throughout the manuscript,* CDKN1A *and* GDF15*, are the two most potently transcribed p53 target genes (Allen et al.,* eLife *2014). However, the authors do not examine the effects that p53 may play in their system. It is entirely possible that most of the effects described upon* MAF1 *knockdown are driven indirectly by activation of p53 downstream of a cellular stress response caused by deregulated RNAPIII function. The authors should check for p53 induction upon* MAF1 *knockdown. Key experiments should be repeated in an isogenic system -/+ p53. This could be done with stable p53 knockdown MCF7 cells (Agami et al.) or HCT116 p53 -/+ cells (Vogelstein et al.)*.

We thank you for your insightful suggestion in bringing p53 to our awareness. Indeed, both *CDKN1A* and *GDF15* are p53 target genes (3) and therefore maybe indirectly activated by p53 instead of *MAF1*. To directly address this question, we perform *MAF1* knockdown experiments in wild-type and p53-null HCT116 cells. Western blot indeed demonstrate the lack of p53 protein in p53-null HCT116 (Figure 1—figure supplement 1; subsection headed “MAF1 knockdown strongly upregulated *CDKN1A* expression”). Since *CDKN1A* expression is induced in both of these cell lines, the induction is independent of p53 (Figure 1—figure supplement 1; same section).

*2) Promoter definition. The authors indicate that ‘Examination of* CDKN1A *gene shows that there are two promoter regions (hereafter referred as long and short form)’. This description contrasts with many earlier reports that clearly show a major promoter for* CDKN1A *with minor alternative start sites. What exactly are the long and short forms? What NM genebank reference numbers do they correspond to? Where are they located relative to the well-characterized p53 enhancers? Importantly, the authors show equivalent Ser5-phospho-RNAPII binding to both promoters, but high H3K4me3 and TBP binding are found only at the downstream promoter encoding the ‘short form’, indicating that this promoter is the functional promoter. Confusingly, although the authors show equivalent binding of Ser5-phospho-RNAPII to both promoters, RBP5, a subunit common to all three polymerases is found preferentially at the downstream promoter. How much RNA is being produced from the ‘long’ and ‘short’ promoters? Some form of quantitative assay should be done to define the relative strength and RNA output of these two ‘promoters’*.

We thank you for highlighting that many earlier reports clearly show a major promoter for *CDKN1A* with minor alternative start sites. Indeed, most published papers refer to the transcription start site (TSS) of short form promoter, whereas the well-characterized p53 enhancers are located 1.4 kb and 2.3 kb upstream from TSS, which is near long form promoter (Hung et al., 2011; Lee et al., 2012). In our manuscript, the long form promoter refers to the upstream region of NM_001220777, whereas the short form promoter refers to the upstream region of NM_001220778 according to NM genebank reference numbers (Figure 3—figure supplement 1). The relative location of the first exons of *CDKN1A* (“L-Ex-1” for long form and “Ex1” for short form) with respect to the identified p53 response elements and ChIP amplicons (p21-L, p21, p2 and p3) is depicted in Figure 9.

Author response image 1.**DOI:**
http://dx.doi.org/10.7554/eLife.06283.016

Therefore, we have modified the text to: “Examination of *CDKN1A* gene in the UCSC Genome Database shows that there are two transcription start sites, NM_001220777 (long form) and NM_001220778 (short form), which are 2.25 kb apart” (Figure 3—figure supplement 1; the subsection headed “*CDKN1A* promoter”). Although the expression of both forms is induced after *MAF1* knockdown, the short form has a higher expression level and stronger promoter activity in MCF-7 cell line (Figure 3—figure supplement 1; same section).

In order to show the relative strength and RNA output of the two promoters, we perform luciferase assay and quantitative RT-PCR to detect the relative promoter activity and transcription level, respectively. In consistent with short form promoter being the major promoter, our luciferase promoter assay showed that the short form *CDKN1A* has a higher promoter activity than the long form (Figure 3—figure supplement 1). Consistent with the above result, our quantitative RT-PCR analysis also shows that the expression of the short form of *CDKN1A* is higher than the long form (Figure 3—figure supplement 1). Since both forms of *CDKN1A* contain active promoters and are transcribed, Ser5-phospho-RNAPII is detected at both promoter regions (Figure 3 and Figure 3—figure supplement 1). However, due to the fact that the short form promoter is the major promoter with higher expression level and promoter activity, the detection of H3K4me3, as well as binding of TBP and RBP5 are found preferentially at short form promoter. The weak binding of these factors on long form promoter is consistent with the lower activity of the long form promoter. However, the biological function of the long form *CDKN1A* with alternative 5′ non-coding exon remains to be elucidated.

*3) Massive misinterpretation of ‘chromatin looping’ data. The authors repeatedly state throughout the manuscript (including the Title) a cause-effect relationship between chromatin looping and gene activation. They state in a subheading of the results that* ‘MAF1 *knockdown enhanced* CDKN1A *gene expression from both long and short form promoters through Pol III-mediated promoter looping’. In the next subheading they repeat:* ‘MAF1 *knockdown enhanced* GDF15 *gene expression through PolIII-mediated promoter looping’. This is a massive misinterpretation of data, because there is no experiment showing a cause-effect relationship between looping and gene activity, only a correlation. The authors state that ‘Moreover, simultaneous knockdown of* MAF1 *with either Pol III or BRF1 (subunit of TFIIIB) disrupted the looping formation (*Figure 5*). These observations suggest chromatin looping between the long and short form promoter regions is important for the activation of* CDKN1A *transcription from both promoters’, and ‘The above results demonstrate that* MAF1 *knockdown can activate CDKN1A expression through promoter looping induced by recruitment of Pol III and Pol II along with histone-modifying factors’. Later on, they state ‘These results further demonstrate unambiguously that* MAF1 *protein represses the promoter activity of* CDKN1A *and* GDF15*, and that recruitment of Pol III after* MAF1 *depletion is crucial for the activation of transcription of these genes through promoter looping’. The looping observed may as well be a consequence, rather than a driver, of gene activation, or just as well a totally inconsequential event. The manuscript should be rewritten to remove all indication of cause-effect relationship between looping and gene activation*.

Indeed, we have no evidence that chromatin looping plays a causal role in the gene expression of the two genes we studied. Therefore, we have rewritten the manuscript to remove the indications of cause-effect relationship between looping and gene activation. We have changed the title to “MAF1-Regulated-Pol III-Mediated Activation of *CDKN1A* and Chromatin Looping”. The subheadings were also modified to “Pol III is required for transcriptional activation and chromatin looping of *CDKN1A*/*GDF15* after *MAF1* knockdown” (in the Results section).

*4) The authors rely on the use of a single cell line, MCF7. As these are breast cancer cells that have sustained multiple genetic alterations, it is unclear whether the findings will hold in normal cells or even in other tumor cell lines. This needs to be addressed as the results with* MAF1 *may represent a peculiar result of the specific genetic make-up of this single, genetically complex cell type*.

To address this question, we choose four other cell lines, MCF-10A (non-tumorigenic), MDA-MB-231, HCT116^p53+/+^ (wild-type) and HCT116^p53-/-^ (p53-null), to repeat our experiments. We performed quantitative RT-PCR to analysis the effect of *MAF1* knockdown and simultaneous knockdown of *MAF1* and *POLR3A* on the expression of *CDKN1A* and *GDF15* in these four cell lines. Consistent with the MCF-7 results, the expression of *CDKN1A* and *GDF15* is induced after *MAF1* knockdown, and simultaneous knockdown of *MAF1* and *POLR3A* abolished the induced expression (Figure 1—figure supplement 1; subsection headed “*MAF1* knockdown strongly upregulated *CDKN1A* expression”). Together, these results demonstrate that *MAF1* regulates *CDKN1A* expression in non-tumorigenic (MCF-10A), as well as breast (MCF-7 and MDA-MB-231) and colon cancer (HCT116^p53+/+^ and HCT116^p53-/-^) cells, and is independent of p53.

*5) The bulk of the findings rely in the use of RNAi approaches but the authors use only a single targeting sequence for the knockdown of each particular target. Multiple targeting sequences need to be used and the extent of downregulation at both the protein and RNA levels needs to be shown to reduce the possibility of off-target effects and to further validate the conclusions*.

To address this question, we ordered two more siRNA targeting sequences for *MAF1* and *POLR3A*. Quantitative RT-PCR and western blot were performed to analyze the knockdown effect of *MAF1* and *POLR3A* using three different siRNAs, including the original one. The results showed that the three siRNAs for *MAF1* depleted *MAF1* and up-regulated tRNA genes, as well as *CDKN1A* and *GDF15* (Figure 1; subsection headed “*MAF1* knockdown strongly upregulated CDKN1A expression”). However, probably due to slow turnover time of *MAF1* protein, western blot analysis showed only 45% of reduction for the *MAF1* protein in cells treated with siRNA (#1, the one used in the original experiment) with the strongest affect. Nevertheless, the reduction percentage for *MAF1* is consistent with the up-regulation of *CDKN1A* in mRNA and protein level, indicating that depletion of *MAF1* induces *CDKN1A* expression (Figure 1; in the aforementioned section). The three siRNAs for *POLR3A* also depleted *POLR3A* and down-regulated tRNA genes (Figure 1—figure supplement 1; subsection headed “*MAF1* knockdown strongly upregulated *CDKN1A* expression”). The siRNAs with the strongest knockdown affect for *MAF1* (#1, the original one) and *POLR3A* (#1, the original one) were used to perform knockdown in ChIP, 3C and reporter assays.

*6) As a general comment both on the cellular knockdown data and the in vitro immuno-depletion data, the authors do not generally indicate the extent of depletion of the factors of interest, and this may well account for some of the modest, albeit significant, effects. Thus it would help to have data/comments on the depletion efficiencies*.

For cellular knockdown, we have performed western blot analysis to show that *MAF1* and *POLR3A* are reduced about 45% and 60% in protein level, respectively (Figure 1; subsection “*MAF1* knockdown strongly upregulated *CDKN1A* expression” and Figure 1—figure supplement 1). For the depletion efficiency of in vitro immune-depletion data, we used anti-Pol II antibody on Pol II genes, *CDKN1A*, *GDF15* and *TAF5*, to show that the transcription activity is reduced 93%±4%, which is similar to treatment with α-amanitin (87%±1%) (Figure 4; subsection “In vitro transcription using HeLa cell nuclear extract demonstrated that transcription of Pol II genes was reciprocally regulated by *MAF1* and Pol III). We also used anti-Pol III antibody on Pol III transcribed-RPPH1 to show that the reduction is 79%±1% (Figure 4; aforementioned subsection). The modest, albeit significant, results may thus be accounted by the depletion level.

*7) In the nuclear run-on assay of*
Figure 1*, the effect of the* MAF1 *knockdown on* CDKN1A *expression appears relatively modest (especially in relation to the clear large effect in the quantitative RT-PCR analysis of*
Figure 1*). Could the authors provide quantitation of the data for* CDKN1A *(and for* GDF15*, where the up-regulation is larger) and perhaps improved data*?

We repeated run-on assay to provide quantification results (n=3) and a more improved gel picture in Figure 2 (please see “*MAF1* knockdown strongly upregulated CDKN1A expression”). The gel pictures were quantified with ImageJ (provided by NIH: http://imagej.nih.gov/ij/). The purified run-on products where normalized with internal control (*GAPDH*) to obtain the relative transcription levels for each gene. *MAF1* knockdown induced the expression of *CDKN1A* and *GDF15* for 2.9±0.8 and 88.8±12.2 folds, respectively, in run-on assay. This shows that *MAF1* knockdown induces newly transcribed *CDKN1A* and *GDF15* mRNA, and the up-regulation is not due to post-transcriptional processing or RNA turnover rates. The run-on results were moved from Figure 1 to Figure 2 for clarity.

*8) The authors state that knocking down Pol III alone did not affect* CDKN1A *expression. However, a close examination the data in*
Figure 1
*suggest that expression of* CDKN1A *is actually reduced about 30% (and that of* GDF15 *more than 50%). This fits the model that Pol III can positively regulate the expression of* CDKN1A *and* GDF15*, but is potentially inconsistent with the authors' failure to see any Pol III occupancy prior to* MAF1 *knockdown. Is this just due to the lower sensitivity and higher background in the ChIP analysis*?

The expression of *CDKN1A* and *GDF15* is indeed reduced about 30% and 50%, respectively, in Pol III knockdown, which fits the model that Pol III can positively regulate the expression of *CDKN1A* and *GDF15*. However, the occupancy of Pol III prior to *MAF1* knockdown is relatively very weak. Thus, we speculate that since the down-regulation of *CDKN1A* and *GDF15* is not as efficient as tRNA genes after Pol III knockdown, the amount of Pol III binding at *CDKN1A* and *GDF15* prior to *MAF1* knockdown may be too low to be differentiated from background (MAF1/Pol III knockdown). Thus, although knockdown Pol III showed a decrease in Pol III binding at tRNA genes, the amount of Pol III is relatively the same as the *CDKN1A* and *GDF15* knockdown control and MAF1/Pol III knockdown. It is possible that there is a minor subpopulation of alleles with Pol III bound at the promoter, and they are responsible for the low level background transcription. Knockdown of Pol III will thus reduce the background expression level from this subset of alleles. However, the low level of binding does not allow us to show that unequivocally. Therefore, we modify the manuscript in the Discussion to state that Pol III may be able to regulate expression of *CDKN1A* and *GDF15*, before *MAF1* knockdown, but the low level of Pol III binding prior to *MAF1* knockdown does not allow us to validate this possibility unequivocally (Discussion).

*9) In*
Figure 3*, the* MAF1 *ChIP signal is only 3-fold above background. Because this data provides important evidence of direct regulation, and even though it is supported by the in vitro demonstration of direct* MAF1 *binding to the SINE element, the authors might consider ways to modify the ChIP protocol to increase the signal/background ratio*.

In order to optimize our ChIP protocol, we performed pilot experiments by adding different concentrations of BSA (0, 5, 10, 20 and 50 μg/ml) during complex binding, as well as using various washing conditions (either with or without LiCl or by washing with different number of times for high-salt washing buffer). The condition used (see Methods) is already our optimized condition after testing various conditions. Our 3-fold signal/background ratio for *MAF1* ChIP is also relatively higher than other published *MAF1* ChIP data, which showed a 2-fold signal/background ratio at Pol II and Pol III promoter in yeast and mammalian cell lines (Graczyk et al., 2011 [Figure 4]; [18] [Figure 7]; Palian et al., 2014 [Figure 5]). Whether the relatively low binding in the MAF1 ChIP assay is due to the intrinsic inefficiency of MAF1 antibody available or to omission of an unknown cofactor remains to be analyzed in the future.

*10) While the in vitro data for the direct MAF1 binding to the p21 SINE element (*Figure 4*) seems convincing, especially with the SINE deletion control, it would be very interesting to have more detailed data on the interaction site in light of the novelty of this observation. Have the authors any further information from DNase footprinting, gel shifts with point mutations in the Pol III promoter elements etc. on this point, especially in relation to their suggestion in the Discussion that “MAF1 and Pol III/TFIIIB may recognize similar DNA elements”? (It should also be noted that the promoter elements of Pol III-transcribed genes are directly recognized by TFIIIC and not by Pol III/IIIB*.*)*

In addition to the SINE deletion control, we constructed point mutations in the A-box Pol III promoter element obtained from *CDKN1A* promoter. We performed in vitro binding assay using the A-box-mutated SINE element as described before (5; 34) (Figure 4; subsection “*MAF1* binds to the short interspersed element (SINE) of *CDKN1A*”). In addition, we also performed in vitro DNA-protein binding assay by using a colorimetric assay kit provided by Abcam (ab117139) to further validate that *MAF1* binds to the Pol III promoter element, but not the mutant form (Figure 4; please also see the aforementioned subsection). We hope that the two assays along with mutant forms of Pol III promoter can provide more detailed data on the interaction site.

*11) In relation to the interesting (and potentially provocative) new* MAF1 *mechanism involving direct binding to promoter elements, the reader is advised (in the Introduction) of prior reports indicating* MAF1 *binding to BRF1 and/or Pol III as well as Pol III/*MAF1 *co-occupancy on target genes, and thus anticipates that* MAF1 *may be repressing Pol III function by a related mechanism on the target genes analyzed here. This, of course, appears not to be the mechanism from the ChIP analyses that follow the functional studies, but I think the rationale and results could be presented with more clarity and with clear indications that, perhaps surprisingly, they do not fit the original models but suggest an alternate mechanism that is pursued (by the direct binding studies)*.

We thank you for highlighting that we should modify the manuscript to present the rationale and results in more clarity and indications that our model does not fit the original models. Therefore, we have rewritten the last paragraph of the Introduction and modified the first paragraph of the Discussion to suggest an alternate mechanism as shown by the direct binding studies.

[Editors’ note: a second round of author responses follow.]

*Despite a commendable effort by the authors to try to address the concerns raised by the reviewers, the revised manuscript fails to meet the high standards of scholarship expected for publication in* eLife*. The authors have performed additional experiments in response to the reviewers suggestions, mostly to include missing controls and replicates, but the manuscript still fails to produce a clear picture of how exactly* MAF1 *and POLIII regulate protein-coding genes such as* CDKN1A *and* GDF15*.*

Overall, this report presents some exciting new findings that are very timely in view of (i) extensive ChIPseq data in the literature demonstrating the localization of components of the Pol III transcription machinery adjacent to Pol II-transcribed genes and (ii) the widespread localization of SINE elements (of unknown function) in the human genome and near promoters”. The decision is based mainly on three criticisms, which we would like to respond to here.

First, PCAF in our manuscript is *not* PCNA as stated in the rejection letter. PCAF is lysine acetyltransferase 2B (an interacting partner of the transcriptional co-integrators CBP and p300), also known as P/CAF, or KAT2B; and it has been shown to be associated with p300 for chromatin modification. We performed ChIP of PCAF to show that this epigenetic factor becomes associated with *CDKN1A* promoter after *MAF1* knockdown (subsection “*MAF1* knockdown promoted recruitment of positive regulatory factors and epigenetic modifications” and Figure 5). Therefore, this is just a misunderstanding.

The second point is about our interpretation of our data in view of the fact the 2.25kb upstream of canonical promoter is probably the enhancer region. We will rewrite the manuscript to incorporate this fact. As Darwin said: “False facts are highly injurious to the progress of science, for they often long endure; but false views, if supported by some evidence, do little harm, as every one takes a salutary pleasure in proving their falseness”. A set of data can always have different interpretation and progress of science is achieved to resolve the different hypothesis or interpretation. We will also perform experiment to test whether the upstream region is the enhancer for the canonical promoter. We will restrict the interpretation in the Discussions section, which is the place used for making hypothesis or interpretation of the data.

The third point is that we failed to correct some points raised by the reviewers such as the test of p53 level by western blot. We did not perform western blot in the revised manuscript because we have already shown that in the revised manuscript, *CDKN1A* and *GDF15* in both HCT116 and HCT116^p53-/-^ cell lines were up-regulated, and thus the effect is independent of p53. Nevertheless, we will perform western blot of p53 to further show that the up-regulation of *CDKN1A* and *GDF15* after *MAF1* knockdown is independent of p53.

*The major issue is that, in response to a request for clarification from the reviewers, it is now evident that the genomic region that the authors have called the ‘p21 long promoter’ (p21L) is indeed the enhancer regulatory region of this gene, known to be bound by p53 and many other transcription factors. This region, located 2.25kb upstream of the canonical p21 promoter, produces enhancer-derived RNAs (eRNAs) and may engage with the promoter through a chromatin loop. In view of this, the entire dataset must be reinterpreted as* MAF1 *and POLIII likely affecting the activity of the enhancer region at the* CDKN1A *locus, and the manuscript should be rewritten in light of this fact*.

The editors questioned the interpretation of our data in view of the fact the 2.25 kb upstream of canonical promoter is probably the enhancer region. For clarification, the TSS start site of NM_001220777 (long form *CDKN1A*) is located 2.25 kb upstream of NM_001220778 (short form *CDKN1A*) TSS. However, the chromatin looping and identified MAF1 binding site by ChIP is 2.65 kb upstream of TSS. Therefore, to address the possibility of enhancer activity at this region, we cloned the 2.25 ∼ 2.65 kb upstream region into the promoter of *CDKN1A* (CDKN1A-S) used in our original luciferase assay. The promoter activity was analyzed by luciferase assay and the results showed that the cloned 5′ region, both in sense (labeled as “LE(+)”) and antisense direction (labeled as “LE(-)”) does not exert enhancer activity on *CDKN1A* promoter after *MAF1* knockdown as shown below (Figure 10). We have included the interpretation in our Discussion section, but did not include the data in the results. In brief, we hypothesized that the chromatin looping we observed for *CDKN1A* and *GDF15* after *MAF1* knockdown may be mediated through the proximity of these 5′ regions in the transcription factory and looping is resulted by the interaction between the sets of transcriptional factors recruited in the two 5′ flanking regions after chromatin remodeling as proposed by [9].

Author response image 2.**DOI:**
http://dx.doi.org/10.7554/eLife.06283.017

*Finally, the revised manuscript failed to correct some points raised by reviewers. As an example of a major oversight, the authors tested for ‘p53 activation’ by Q-RT-PCR, when it is well established that p53 activation upon cellular stress occurs at the protein level, and thus should have been tested by Western blot*.

The other concern raised by the editors is that we failed to test p53 level by western blot to determine whether the induced *CDKN1A* expression after *MAF1* knockdown is independent or dependent of p53. To directly show that induced *CDKN1A* expression after *MAF1* knockdown is independent of p53, we analyzed the *MAF1* knockdown effect of *CDKN1A* expression in p53-null HCT116. Western blot analysis indeed confirmed that this cell line does not express p53 at protein level (Figure 1—figure supplement 1; subsection entitled “*MAF1* knockdown strongly upregulated *CDKN1A* expression”). Both quantitative RT-PCR and western blot showed that the expression of *CDKN1A* is induced after *MAF1* knockdown in p53-null HCT116 cell line (Figure 1—figure supplement 1; aforementioned subsection). This result unequivocally shows that in the absence of p53 protein, the expression of *CDKN1A* is still induced after *MAF1* knockdown, indicating the induced transcription is independent of p53. Consistent with this result, we also showed that induction of *CDKN1A* expression in MDA-MB-231 cell line, which carries mutation in p53 (Figure 1—figure supplement 1; same subsection).